# Theoretical Analysis on the Generalization Power of Overfitted Transfer Learning

## Abstract

Transfer learning is a useful technique for achieving improved performance and reducing training costs by leveraging the knowledge gained from source tasks and applying it to target tasks. Assessing the effectiveness of transfer learning relies on understanding the similarity between the ground truth of the source and target tasks. In real-world applications, tasks often exhibit partial similarity, where certain aspects are similar while others are different or irrelevant. To investigate the impact of partial similarity on transfer learning performance, we focus on a linear regression model with two distinct sets of features: a common part shared across tasks and a task-specific part. Our study explores various types of transfer learning, encompassing two options for parameter transfer. By establishing a theoretical characterization on the error of the learned model, we compare these transfer learning options, particularly examining how generalization performance changes with the number of features/parameters in both underparameterized and overparameterized regimes. Furthermore, we provide practical guidelines for determining the number of features in the common and task-specific parts for improved generalization performance. For example, when the total number of features in the source task's learning model is fixed, we show that it is more advantageous to allocate a greater number of redundant features to the task-specific part rather than the common part. Moreover, in specific scenarios, particularly those characterized by high noise levels and small true parameters, sacrificing certain *true* features in the common part in favor of employing more *redundant* features in the task-specific part can yield notable benefits.

## 1 Introduction

Transfer learning is a powerful technique that enhances the learning performance of a target task by leveraging knowledge from a related source task (Pan & Yang, 2010). There are two main categories of transfer learning: parameter transfer and sample transfer. In parameter transfer, the learned parameters from the source task are directly copied to the target task's learning model. In sample transfer, training samples from the source task are integrated into the target task's dataset and contribute to its training process. Comparing these two methods, sample transfer can provide additional valuable information and allow for preprocessing of the transferred samples to better align them with the target task, while parameter transfer offers significant savings in training costs and thus is very helpful for models with a large number of parameters such as deep neural networks (DNNs).

Despite the proven effectiveness of transfer learning with DNNs in various real-world applications, a comprehensive theoretical understanding of its performance remains under-explored. DNNs are typically overparameterized, allowing them to fit all training samples while maintaining relatively good generalization performance. This behavior challenges our understanding of the classical bias-variance trade-off. Recent studies have explored the phenomenon of "double-descent" or "benign overfitting" in certain linear regression setups, where the test error descends again in the overparameterized region, shedding light on this mystery. However, most of the existing literature focuses on single-task learning. The existence of a similar phenomenon in transfer learning, even in the simple linear regression setting, remains insufficiently explored. The additional transfer process in transfer learning makes the analysis of the generalization performance in the underparameterized and overparameterized regimes considerably more complex. Furthermore, quantifying task similarity

necessitates the development of appropriate analytical methods to establish a connection with the generalization performance of transfer learning.

**The contribution of this paper is as follows.** In this paper, we investigate the generalization performance of transfer learning in linear regression models under both the underparameterized and overparameterized regimes. Compared to the existing literature that considers a general noisy linear relation between the true parameters of the source and target tasks, we delve into the separation between common and task-specific features in greater detail. Specifically, we partition the feature space into a common part and a task-specific part. This setup enables us to analyze how the number of parameters in different parts influences the generalization performance of the target task. By characterizing the generalization performance, we offer insightful findings on transfer learning. For instance, when the total number of features in the source task's learning model is fixed, our analysis reveals the advantage of *allocating more redundant features to the task-specific part rather than the common part*. Additionally, in specific scenarios characterized by high noise levels and small true parameters, *sacrificing certain true features in the common part in favor of employing more redundant features in the task-specific part can yield notable benefits*.

## 1.1 RELATED WORK

"Benign overfitting" and "double-descent" have been discovered and studied for overfitted solutions in single-task linear regression. Some works have explored double-descent with minimum $\ell_2$-norm overfitted solutions (Belkin et al., 2018; 2019; Bartlett et al., 2020; Hastie et al., 2019; Muthukumar et al., 2019) or minimum $\ell_1$-norm overfitted solutions (Mitra, 2019; Ju et al., 2020), while employing simple features such as Gaussian or Fourier features. In recent years, other studies have investigated overfitted generalization performance by utilizing features that approximate shallow neural networks. For example, researchers have explored random feature (RF) models (Mei & Montanari, 2019), two-layer neural tangent kernel (NTK) models (Arora et al., 2019; Satpathi & Srikant, 2021; Ju et al., 2021), and three-layer NTK models (Ju et al., 2022). Note that all of these studies have focused solely on a single task.

There are only a limited number of studies on the theoretical analysis of transfer learning. Lampinen & Ganguli (2019) investigate the generalization dynamics in transfer learning by multilayer linear networks using a student-teacher scenario where the teacher network generates data for the student network, which is different from our setup where the data of the source task and the target task are independently generated by their own ground truth. Dhifallah & Lu (2021) focus on the problem of when transfer learning is beneficial using the model of the single-layer perceptron. Gerace et al. (2022) study a binary classification problem by transfer learning of the first layer in a two-layer neural network. However, both Dhifallah & Lu (2021) and Gerace et al. (2022) include an explicit regularization term in their models, which prevents overfitting. There are also some recent studies of transfer learning on linear models (Bastani, 2021; Li et al., 2022; Tian & Feng, 2022; Li et al., 2023; Tripuraneni et al., 2020; Zhang et al., 2022; Lin & Reimherr, 2022). For example, Bastani (2021) and Li et al. (2022) investigate estimation and prediction in high-dimensional linear models. Tian & Feng (2022) and Li et al. (2023) further extend the setup to high-dimensional generalized linear models. Tripuraneni et al. (2020) consider the case where source and target tasks share a common and low-dimensional linear representation. Lin & Reimherr (2022) study transfer learning in a functional linear regression where the similarity between source and target tasks is measured using the Reproducing Kernel Hilbert Spaces norm. Zhang et al. (2022) provide minimax bounds on the generalization performance but do not overfit the training data. In particular, none of these studies have considered the task similarity structure of interest in this paper, nor investigated the generalization performance in both overparameterized and underparameterized regimes.

The most related work to ours is Dar & Baraniuk (2022). Specifically, Dar & Baraniuk (2022) studies the double descent phenomenon in transfer learning, which is also our focus in this paper. However, Dar & Baraniuk (2022) does not consider an explicit separation of the feature space by the common part and the task-specific part like we do in this paper. As we will show, such a separation in the system model enables us to analyze the double descent phenomenon under different options for transfer learning, including two options for parameter transfer and two options for data transfer. In contrast, Dar & Baraniuk (2022) only studies one option of parameter transfer. Therefore, our analysis is quite different from that of Dar & Baraniuk (2022).

## 2 System Model

### 2.1 Linear Ground truth involving multiple tasks

In a classical single-task linear regression, ground truth parameters are treated as one vector, and all corresponding features (each feature is a scalar) are also treated as one vector. However, when involving multiple tasks, due to the partial similarity among different tasks, using only one vector to represent the ground truth parameters and features is no longer enough. A finer linear model should consider the common part and the task-specific part separately. Here we consider one training (source) task and one test (target) task, respectively referred to as the first and second task from now on. We consider a linear model for each task; i.e., for the $i$-th task with $i \in \{1 \text{ (source)}, 2 \text{ (target)}\}$, samples are generated by

$$y_{(i)} = \hat{\boldsymbol{x}}^T \hat{\boldsymbol{w}}_{(i)} + \hat{\boldsymbol{z}}_{(i)}^T \hat{\boldsymbol{q}}_{(i)} + \epsilon_{(i)}, \tag{1}$$

where $\hat{\boldsymbol{x}} \in \mathbb{R}^s$ denotes the value of the features that correspond to the similar/common parameters $\hat{\boldsymbol{w}}_{(i)} \in \mathbb{R}^s$, $\hat{\boldsymbol{z}}_{(i)} \in \mathbb{R}^{s_{(i)}}$ denotes the value of the features that correspond to the task-specific parts $\hat{\boldsymbol{q}}_{(i)} \in \mathbb{R}^{s_{(i)}}$, and $\epsilon_{(i)} \in \mathbb{R}$ denotes the noise. Here, $s$ denotes the number of common features and $s_{(i)}$ denotes the number of $i$-th task-specific features. Let $\hat{\mathcal{S}}_{(i)}$ denote the set of features corresponding to $\hat{\boldsymbol{z}}_{(i)}$ and $\hat{\mathcal{S}}_{\text{co}}$ the set of features corresponding to $\hat{\boldsymbol{x}}$ (so their cardinality $\left|\hat{\mathcal{S}}_{(i)}\right| = s_i$ and $\left|\hat{\mathcal{S}}_{\text{co}}\right| = s$).

**Representative motivating example:** In real-world applications, many tasks actually have such a partial similarity structure. For example, for image recognition tasks, some low-level features are common (e.g., skin texture of animals, surface of a machine) among different tasks even if the objectives of those tasks are completely different (e.g., classifying cat and airplane, or classifying dog and automobile). These low-level features are usually captured by convolutional layers in DNNs, while the remaining parts of the DNNs (e.g., full-connected layers) are used to extract task-specific features. Even for a simple linear regression model, a theoretical explanation of the effect of common features and task-specific features on the generalization performance of transfer learning may provide useful insights on designing more suitable real-world transfer learning model structures (e.g., how many neurons to use in convolutional layers of DNNs to extract common low-level features to transfer).

### 2.2 Feature selection for learning

From the learner's point of view, the true feature sets $\hat{\mathcal{S}}_{\text{co}}$ and $\hat{\mathcal{S}}_{(i)}$ are usually unknown for many real-world applications. In the overparameterized regime, redundant parameters (along with redundant features) are used/selected more than necessary, which is characterized by the following assumption. Choosing redundant features also means that the learner does not need to be very precise in distinguishing the common and task-specific features, since the learner can include "suspicious" features in the common feature set.

**Definition 1.** $\hat{\mathcal{S}}_{co} \subseteq \mathcal{S}_{co}$ and $\hat{\mathcal{S}}_{(i)} \subseteq \mathcal{S}_{(i)}$ for all $i \in \{1, 2\}$, where $\mathcal{S}_{co}$ denotes the set of selected features for the common part, and $\mathcal{S}_{(i)}$ denotes the set of selected task-specific features.

Define $p := |\mathcal{S}_{\text{co}}|$ and $p_{(i)} := \left|\mathcal{S}_{(i)}\right|$. Let $\tilde{\boldsymbol{w}} \in \mathbb{R}^p$ denote the parameters to learn the common part and $\tilde{\boldsymbol{q}}_{(i)} \in \mathbb{R}^{p_{(i)}}$ the parameters to learn the $i$-th task's specific part.

With Definition 1, we construct $\boldsymbol{w}_{(i)} \in \mathbb{R}^p$ (corresponding to $\mathcal{S}_{\text{co}}$) from $\hat{\boldsymbol{w}}_{(i)}$ (corresponding to $\hat{\mathcal{S}}_{\text{co}}$) by filling zeros in the positions of the redundant features (corresponding to $\mathcal{S}_{\text{co}} \setminus \hat{\mathcal{S}}_{\text{co}}$). We similarly construct $\boldsymbol{q}_{(i)} \in \mathbb{R}^{p_{(i)}}$ from $\hat{\boldsymbol{q}}_{(i)}$. Thus, Eq. (1) can be alternatively expressed as

$$y_{(i)} = \boldsymbol{x}^T \boldsymbol{w}_{(i)} + \boldsymbol{z}_{(i)}^T \boldsymbol{q}_{(i)} + \epsilon_{(i)}, \tag{2}$$

where $\boldsymbol{x} \in \mathbb{R}^p$ are the features of $\hat{\mathcal{S}}_{\text{co}}$ and $\boldsymbol{z}_{(i)} \in \mathbb{R}^{p_{(i)}}$ are the features of $\mathcal{S}_{(i)}$. Notice that the ground truth (i.e., input and output) does not change with $p$ or $p_{(i)}$ (since it only changes how many additional zeros are added).

For analytical tractability, we adopt Gaussian features and noise, which is formally stated by the following assumption.

**Assumption 1.** *All features follow* i.i.d.[1] *standard Gaussian $\mathcal{N}(0, 1)$. The noise also follows the Gaussian distribution. Specifically, $\epsilon_{(1)} \sim \mathcal{N}\left(0, \sigma_{(1)}^2\right)$ and $\epsilon_{(2)} \sim \mathcal{N}\left(0, \sigma_{(2)}^2\right)$.*

**Remark 1.** *If there exist some missing features[2] in $\mathcal{S}_{co}$ and $\mathcal{S}_{(i)}$ (i.e., Definition 1 is not satisfied), then the effect of these missing features is the same as the noise since we adopt* i.i.d. *Gaussian features. Thus, our methods and results still hold by redefining $\sigma_{(1)}^2$ and $\sigma_{(2)}^2$ as the total power of the noise and the missing features, i.e., $\sigma_{(i)}^2 \leftarrow \sigma_{(i)}^2 + \left\|\hat{\boldsymbol{w}}_{(i)}^{missing}\right\|^2 + \left\|\hat{\boldsymbol{q}}_{(i)}^{missing}\right\|^2$ where $\hat{\boldsymbol{w}}_{(i)}^{missing}$ and $\hat{\boldsymbol{q}}_{(i)}^{missing}$ denote the sub-vectors for the missing features of $\hat{\boldsymbol{w}}_{(i)}$ and $\hat{\boldsymbol{q}}_{(i)}$, respectively.*

## 2.3 TRAINING SAMPLES AND TRAINING LOSSES

Let $n_{(i)}$ denote the number of training samples for task $i \in \{1, 2\}$. We stack these $n_{(i)}$ samples as matrices/vectors $\mathbf{X}_{(i)} \in \mathbb{R}^{p \times n_{(i)}}$, $\mathbf{Z}_{(i)} \in \mathbb{R}^{p_{(i)} \times n_{(i)}}$, $\boldsymbol{y}_{(i)} \in \mathbb{R}^{n_{(i)}}$, where the $j$-th column of $\mathbf{X}_{(i)}$, the $j$-th column of $\mathbf{Z}_{(i)}$, and the $j$-th element of $\boldsymbol{y}_{(i)}$ correspond to $(\boldsymbol{x}, \boldsymbol{z}_{(i)}, y_{(i)})$ in Eq. (2) of the $j$-th training sample. Now Eq. (2) can be written into a matrix equation for training samples:

$$\boldsymbol{y}_{(i)} = \mathbf{X}_{(i)}^T \boldsymbol{w}_{(i)} + \mathbf{Z}_{(i)}^T \boldsymbol{q}_{(i)} + \boldsymbol{\epsilon}_{(i)}, \tag{3}$$

where $\boldsymbol{\epsilon}_{(i)} \in \mathbb{R}^{n_{(i)}}$ is the stacked vector that consists of the noise in the output of each training sample (i.e., $\epsilon_{(i)}$ in Eq. (2)).

We use mean squared error (MSE) as the training loss for the $i$-th task with the learner's parameters $\bar{\boldsymbol{w}}, \bar{\boldsymbol{q}}$ as: $\quad \mathcal{L}_{(i)}^{\text{train}}(\bar{\boldsymbol{w}}, \bar{\boldsymbol{q}}) \coloneqq \frac{1}{n_{(i)}} \left\|\boldsymbol{y}_{(i)} - \mathbf{X}_{(i)}^T \bar{\boldsymbol{w}} - \mathbf{Z}_{(i)}^T \bar{\boldsymbol{q}}\right\|^2$.

## 2.4 OPTIONS OF PARAMETER TRANSFER

The process of transfer learning by transferring parameters consists of three steps: **step 1**) train for the source task using samples $(\mathbf{X}_{(1)}, \mathbf{Z}_{(1)}; \boldsymbol{y}_{(1)})$; **step 2**) select the parameters for the common features $\mathcal{S}_{co}$ from the learned result of the source task and then send them to the target task model; and **step 3**) determine/train the parameters for the target task using its own samples $(\mathbf{X}_{(2)}, \mathbf{Z}_{(2)}; \boldsymbol{y}_{(2)})$ based on the transferred parameters in step 2.

Step 1 is similar to a classical single-task linear regression. The training process will converge to a solution $\tilde{\boldsymbol{w}}_{(1)}, \tilde{\boldsymbol{q}}_{(1)}$ that minimizes this training loss, i.e., $(\tilde{\boldsymbol{w}}_{(1)}, \tilde{\boldsymbol{q}}_{(1)}) \coloneqq \arg\min_{\bar{\boldsymbol{w}}, \bar{\boldsymbol{q}}} \mathcal{L}_{(1)}^{\text{train}}(\bar{\boldsymbol{w}}, \bar{\boldsymbol{q}})$. When $p + p_{(1)} > n_{(1)}$ (overparameterized), there exist multiple solutions that can make the training loss zero (with probability 1). In this situation, we will choose the one with the smallest $\ell_2$-norm $(\tilde{\boldsymbol{w}}_{(1)}, \tilde{\boldsymbol{q}}_{(1)})$ which is defined as the solution of the following optimization problem: $\min_{\bar{\boldsymbol{w}}, \bar{\boldsymbol{q}}} \quad \|\bar{\boldsymbol{w}}\|^2 + \|\bar{\boldsymbol{q}}\|^2 \quad$ subject to $\quad \mathbf{X}_{(1)}^T \bar{\boldsymbol{w}} + \mathbf{Z}_{(1)}^T \bar{\boldsymbol{q}} = \boldsymbol{y}_{(1)}$. We are interested in this minimum $\ell_2$-norm solution among all overfitted solutions because it corresponds to the convergence point of stochastic gradient descent (SGD) or gradient descent (GD) training with zero initial point (see proof in Lemma 5).

Steps 2 and 3 jointly determine the learned result for the target task $\tilde{\boldsymbol{w}}_{(2)}$ and $\tilde{\boldsymbol{q}}_{(2)}$. In this paper, we analyze two possible options differentiated by the usage of the transferred common part $\tilde{\boldsymbol{w}}_{(1)}$.

**Option A (Transfer and Fix):** We directly copy the learned result, i.e., $\tilde{\boldsymbol{w}}_{(2)} \coloneqq \tilde{\boldsymbol{w}}_{(1)}$. For the training of the target task, only the task-specific parameters are trained. In other words, $\tilde{\boldsymbol{q}}_{(2)} \coloneqq \arg\min_{\bar{\boldsymbol{q}}} \mathcal{L}_{(2)}^{\text{train}}(\bar{\boldsymbol{w}}, \bar{\boldsymbol{q}})$ when underparameterized. When $p_{(i)} > n_{(2)}$ (overparameterized), there exist multiple solutions that can make the training loss zero. We then define $\tilde{\boldsymbol{q}}_{(2)}$ as the minimum $\ell_2$-norm overfitted solution, i.e., $\tilde{\boldsymbol{q}}_{(2)}$ is defined as the solution of the following optimization problem: $\min_{\bar{\boldsymbol{q}}} \|\bar{\boldsymbol{q}}\|^2 \quad$ subject to $\quad \mathbf{X}_{(2)}^T \tilde{\boldsymbol{w}}_{(1)} + \mathbf{Z}_{(2)}^T \tilde{\boldsymbol{q}}_{(2)} = \boldsymbol{y}_{(2)}$.

**Option B (Transfer and Train):** We only use the learned common part as an initial training point of $\tilde{\boldsymbol{w}}_{(2)}$. In this option, both $\tilde{\boldsymbol{w}}_{(2)}$ and $\tilde{\boldsymbol{q}}_{(2)}$ are determined by the training of the source task. Specifically, $(\tilde{\boldsymbol{w}}_{(2)}, \tilde{\boldsymbol{q}}_{(2)}) \coloneqq \arg\min_{\bar{\boldsymbol{w}}, \bar{\boldsymbol{q}}} \mathcal{L}_{(2)}^{\text{train}}(\bar{\boldsymbol{w}}, \bar{\boldsymbol{q}})$ when underparameterized. When $p + p_{(2)} > n_{(2)}$,

---

[1]In Appendix F, we numerically check our results and insights in the situation of non-*i.i.d.* settings.
[2]A missing feature means that a true feature is not included in the data.

there are multiple solutions that can make $\mathcal{L}_{(2)}(\bar{\boldsymbol{w}}, \bar{\boldsymbol{q}}) = 0$. We then define $(\tilde{\boldsymbol{w}}_{(2)}, \tilde{\boldsymbol{q}}_{(2)})$ as the convergence point of SGD/GD starting from $(\bar{\boldsymbol{w}} = \tilde{\boldsymbol{w}}_{(1)}, \bar{\boldsymbol{q}} = \mathbf{0})$. Indeed, $(\tilde{\boldsymbol{w}}_{(2)}, \tilde{\boldsymbol{q}}_{(2)})$ corresponds to the smallest $\ell_2$-norm of the difference between the result and the initial point (see proof in Lemma 5):

$$\min_{\bar{\boldsymbol{w}}, \bar{\boldsymbol{q}}} \quad \left\| \bar{\boldsymbol{w}} - \tilde{\boldsymbol{w}}_{(1)} \right\|^2 + \left\| \bar{\boldsymbol{q}} \right\|^2 \quad \text{subject to} \quad \mathbf{X}_{(2)}^T \bar{\boldsymbol{w}} + \mathbf{Z}_{(2)}^T \bar{\boldsymbol{q}} = \boldsymbol{y}_{(2)}.$$

## 2.5 Performance evaluation

We define the *model error* for the target task as

$$\mathcal{L} := \left\| \tilde{\boldsymbol{w}}_{(2)} - \boldsymbol{w}_{(2)} \right\|^2 + \left\| \tilde{\boldsymbol{q}}_{(2)} - \boldsymbol{q}_{(2)} \right\|^2. \tag{4}$$

It can be proven that the model error $\mathcal{L}$ is the expected test loss on noiseless test samples. To make our results in the following sections concise, we define

$$\mathcal{L}_{\text{co}} := \mathop{\mathbb{E}}_{\mathbf{X}_{(1)}, \mathbf{Z}_{(1)}, \boldsymbol{\epsilon}_{(1)}} \left\| \boldsymbol{w}_{(2)} - \tilde{\boldsymbol{w}}_{(1)} \right\|^2 \qquad \text{(transferring error)}, \tag{5}$$

$$\mathcal{L}_{\text{co}}^{\text{noiseless}} := \mathcal{L}_{\text{co}}\big|_{\sigma_{(1)} = \mathbf{0}} \qquad \text{(transferring error when } \sigma_{(1)} = \mathbf{0}), \tag{6}$$

$$\delta := \left\| \boldsymbol{w}_{(2)} - \boldsymbol{w}_{(1)} \right\| \qquad \text{(similarity on common features)}, \tag{7}$$

$$r := 1 - \frac{n_{(1)}}{p + p_{(1)}} \qquad \text{(overparameterized ratio in step 1)}.$$

Intuitively, $\mathcal{L}_{\text{co}}$ describes how well the common part learned from the source task estimates the target task's common part, $\delta$ reflects the similarity between the common parts of the source task and the target task, and $r$ can be regarded as the overparameterization ratio in step 1 introduced in Section 2.4.

## 3 Main Results for Parameter Transfer

For the scheme of transferring parameters (Section 2.4), we will establish three theorems corresponding to the performance of the transferring error[3], the model error of Option A, and the model error of Option B, respectively.

**Theorem 1** (transferring error). *The transferring error (defined in Eq. (5)) is given by*

$$\mathcal{L}_{co} = \begin{cases} \mathcal{L}_{co}^{noiseless} + b_{noise}, & \text{for } p + p_{(1)} > n_{(1)} + 1, \tag{8} \\ \delta^2 + \underbrace{\dfrac{p\sigma_{(1)}^2}{n_{(1)} - (p + p_{(1)}) - 1}}_{\text{Term O1}}, & \text{for } n_{(1)} > p + p_{(1)} + 1, \tag{9} \end{cases}$$

*where $0 \le \mathcal{L}_{co}^{noiseless} \le \min_{i=1,2,3} \bar{b}_i^2$, and*

$$\bar{b}_1 := \delta + \sqrt{r \left( \left\| \boldsymbol{w}_{(1)} \right\|^2 + \left\| \boldsymbol{q}_{(1)} \right\|^2 \right)}, \tag{10}$$

$$\bar{b}_2 := \left\| \boldsymbol{w}_{(2)} \right\| + \sqrt{1 - r} \left\| \boldsymbol{w}_{(1)} \right\| + \sqrt{\min\{r, 1 - r\}} \left\| \boldsymbol{q}_{(1)} \right\|, \tag{11}$$

$$\bar{b}_3 := \sqrt{r} \left\| \boldsymbol{w}_{(1)} \right\| + \delta + \sqrt{\min\{r, 1 - r\}} \left\| \boldsymbol{q}_{(1)} \right\|, \tag{12}$$

$$b_{noise} := \frac{p}{p + p_{(1)}} \cdot \frac{n_{(1)} \sigma_{(1)}^2}{p + p_{(1)} - n_{(1)} - 1}. \tag{13}$$

**Theorem 2** (Option A). *For Option A, we must have*

$$\mathbb{E}[\mathcal{L}] = \begin{cases} \underbrace{\mathcal{L}_{co} + \dfrac{n_{(2)} \left( \mathcal{L}_{co} + \sigma_{(2)}^2 \right)}{p_{(2)} - n_{(2)} - 1}}_{\text{Term A1}} + \underbrace{\left( 1 - \dfrac{n_{(2)}}{p_{(2)}} \right) \left\| \boldsymbol{q}_{(2)} \right\|^2}_{\text{Term A2}}, & \text{for } p_{(2)} > n_{(2)} + 1, \tag{14} \\[2em] \mathcal{L}_{co} + \dfrac{p_{(2)} \left( \mathcal{L}_{co} + \sigma_{(2)}^2 \right)}{n_{(2)} - p_{(2)} - 1}, & \text{for } n_{(2)} > p_{(2)} + 1. \tag{15} \end{cases}$$

---

[3]The error caused by the transferred parameters. The precise definition is given in Eq. (5).

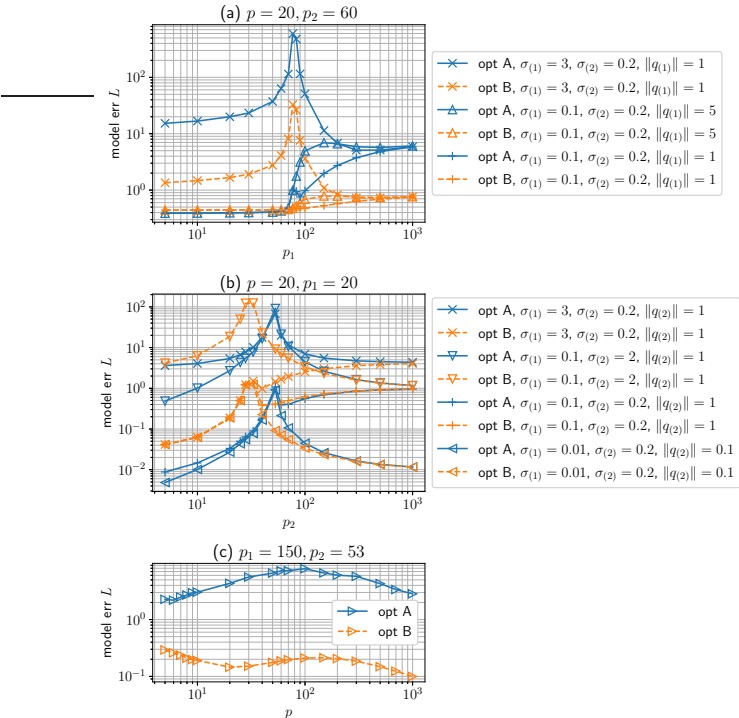

Figure 1: Generalization performance of transfer learning under different setups, where $s = s_1 = s_2 = 5$, $n_{(1)} = 100$, $n_{(2)} = 50$, $\boldsymbol{w}_{(1)} = \boldsymbol{w}_{(2)}$. Each point is the average of 100 random runs. Other settings for each subfigure: (a) $\left\|\boldsymbol{q}_{(2)}\right\| = \left\|\boldsymbol{w}_{(1)}\right\| = 1$; (b) $\left\|\boldsymbol{w}_{(1)}\right\| = \left\|\boldsymbol{q}_{(1)}\right\| = 1$; (c) $\left\|\sigma_{(1)}\right\| = 1$, $\left\|\sigma_{(2)}\right\| = 0.2$, $\left\|\boldsymbol{q}_{(1)}\right\| = \left\|\boldsymbol{q}_{(2)}\right\| = \left\|\boldsymbol{w}_{(1)}\right\| = 0.1$.

**Theorem 3** (Option B). *For Option B, we must have*

$$
\mathbb{E}[\mathcal{L}] = \begin{cases} \underbrace{\left(1 - \frac{n_{(2)}}{p+p_{(2)}}\right)\left(\mathcal{L}_{co} + \left\|\boldsymbol{q}_{(2)}\right\|^2\right)}_{\text{Term B1}} + \underbrace{\frac{n_{(2)}\sigma_{(2)}^2}{p+p_{(2)}-n_{(2)}-1}}_{\text{Term B2}}, & \text{for } p + p_{(2)} > n_{(2)} + 1, \quad (16) \\[2em] \dfrac{(p+p_{(2)})\sigma_{(2)}^2}{n_{(2)} - (p+p_{(2)}) - 1}, & \text{for } n_{(2)} > p + p_{(2)} + 1. \quad (17) \end{cases}
$$

The proofs of Theorems 1 to 3 are given in Appendices B to D, respectively. Theorems 1 to 3 provide some interesting insights, which we now discuss in Sections 3.1 to 3.3.

## 3.1 COMMON INSIGHTS FOR OPTIONS A AND B

**(1) Benign overfitting[4] w.r.t. $p_{(1)}$ needs large $\sigma_{(1)}$.** For the overparameterized regime result in Eq. (8) of Theorem 1, when $\sigma_{(1)}$ is large, the term $b_{\text{noise}}$ (defined in Eq. (13)) dominates $\mathcal{L}_{co}$ and is monotone decreasing w.r.t. $p_{(1)}$. When $p_{(1)} \to \infty$, we have $b_{\text{noise}} \to 0$. In contrast, for the underparameterized regime result in Eq. (9), Term O1 (noise effect) is always larger than $\frac{p\sigma_{(1)}^2}{n_{(1)}}$, which can be worse than that of the overparameterized regime when $p_{(1)}$ is sufficiently large. By Theorems 2 and 3, we know that $\mathcal{L}$ decreases when $\mathcal{L}_{co}$ decreases. Therefore, in the situation of large $\sigma_{(1)}$, increasing $p_{(1)}$ in the overparameterized regime (of step 1) w.r.t. $p_{(1)}$ can reduce the generalization error, which implies the existence of benign overfitting.

We also numerically verify the impact of $\sigma_{(1)}$ on the benign overfitting in Fig. 1(a), where we plot the empirical average of $\mathcal{L}$ w.r.t. $p_{(1)}$. The two curves of $\sigma_{(1)} = 3$ with markers "×" descend in the

---

[4]i.e., the test error of the overparameterized regime is lower than that of the underparameterized regime.

overparameterized regime ($p_{(1)} > 80$) and can be lower than their values in the underparameterized regime. In contrast, the two curves of $\sigma_{(1)} = 0.1$ with markers "+" increase in most parts of the overparameterized regime and are higher than the underparameterized regime. Such a contrast indicates the benign overfitting w.r.t. $p_{(1)}$ needs large $\sigma_{(1)}$.

**(2) Benign overfitting w.r.t. $p_{(2)}$ needs large $\sigma_{(2)}$.** For Eq. (15) (underparameterized regime of Option A), $\mathbb{E}[\mathcal{L}]$ is always larger than $\mathcal{L}_{\text{co}}(1 + \frac{1}{n_{(2)}})$. In contrast, for Eq. (14) (overparameterized regime of Option A), when $\sigma_{(2)}$ is much larger than $\left\| \boldsymbol{q}_{(2)} \right\|^2$, then Term A2 is negligible and Term A1 dominates. In this situation, $\mathbb{E}[\mathcal{L}]$ is monotone decreasing w.r.t. $p_{(2)}$ and will approach $\mathcal{L}_{\text{co}}$ when $p_{(2)} \to \infty$. In other words, benign overfitting exists. Similarly, by Theorem 3, benign overfitting exists when $\sigma_{(2)}^2$ is much larger than $\mathcal{L}_{\text{co}} + \left\| \boldsymbol{q}_{(2)} \right\|^2$.

In Fig. 1(b), the two curves with markers "$\triangledown$" denote the model error of Option A and Option B when $\sigma_{(2)}$ is large ($\sigma_{(2)} = 2$). They have a descending trend in the entire overparameterized regime. In contrast, the two curves with markers "+", which denote the model error for the situation of small $\sigma_{(2)}$ ($\sigma_{(2)} = 0.2$), only decrease w.r.t. $p_{(2)}$ at the beginning of the overparameterized regime, while increasing thereafter.

**(3) A descent floor[5] w.r.t. $p_{(2)}$ sometimes exists.** For Eq. (14) of Option A, Term A1 is monotone decreasing w.r.t. $p_{(2)}$, while Term A2 is monotone increasing w.r.t. $p_{(2)}$. When $p_{(2)}$ is a little larger than $n_{(2)}$, the denominator $p_{(2)} - n_{(2)} - 1$ in Term A1 is close to zero, and thus Term A1 dominates and causes $\mathbb{E}[\mathcal{L}]$ to be decreasing w.r.t. $p_{(2)}$. When $p_{(2)}$ gradually increases to infinity, $\mathbb{E}[\mathcal{L}]$ will approach $\mathcal{L}_{\text{co}} + \left\| \boldsymbol{q}_{(2)} \right\|^2$. By calculating $\partial \mathbb{E}[\mathcal{L}]/\partial p_{(2)}$, we can tell that if $\mathcal{L}_{\text{co}} + \sigma_{(2)}^2 < \left\| \boldsymbol{q}_{(2)} \right\|^2$, in the overparameterized regime, $\mathbb{E}[\mathcal{L}]$ will first decrease and then increase, which implies a descent floor (by Lemma 9 in Appendix A.1). Similarly, by calculating $\partial \mathbb{E}[\mathcal{L}]/\partial p_{(2)}$ for Eq. (16) of Option B, if $\sigma_{(2)}^2 < \mathcal{L}_{\text{co}} + \left\| \boldsymbol{q}_{(2)} \right\|^2$, in the overparameterized regime, $\mathbb{E}[\mathcal{L}]$ will have a descent floor w.r.t. $p_{(2)}$ (by Lemma 10 in Appendix A.1). An interesting observation related to the descent floor is that *the condition of the existence of the descent floor is different for Option A and Option B, where Option A needs small $\mathcal{L}_{co}$ but Option B needs large $\mathcal{L}_{co}$.*

In Fig. 1(b), we see that both curves with markers "+" have a descent floor in the overparameterized regime. In contrast, for the two curves with markers "$\times$" where $\sigma_{(1)}$ is large, only Option B has a descent floor while Option A does not. Since large $\sigma_{(1)}$ implies large $\mathcal{L}_{\text{co}}$, such a difference confirms that the descent floor of Option A needs small $\mathcal{L}_{\text{co}}$ while the one of Option B needs large $\mathcal{L}_{\text{co}}$.

**(4) The effect of $\boldsymbol{q}_{(1)}$ is negligible when heavily or slightly overparameterized in step 1.** The effect of $\boldsymbol{q}_{(1)}$ on $\mathcal{L}$ is through $\mathcal{L}_{\text{co}}^{\text{noiseless}}$. By Eqs. (8) and (10) to (12), the coefficient of $\left\| \boldsymbol{q}_{(2)} \right\|$ is $\min\{r, 1 - r\}$. When heavily overparameterized in step 1, we have $p + p_{(1)} \gg n_{(1)}$ and thus $r \approx 0$. When slightly overparameterized in step 1, we have $p + p_{(1)} \approx n_{(1)}$ and thus $r \approx 1$. In both situations, we have the coefficient $\min\{r, 1 - r\} \approx 0$, which implies that the effect of $\boldsymbol{q}_{(1)}$ is negligible when heavily or slightly overparameterized in step 1.

In Fig. 1(a), we compare two curves with markers "$\triangle$" (for large $\boldsymbol{q}_{(1)}$ that $\left\| \boldsymbol{q}_{(1)} \right\| = 5$) against two curves with markers "+" (for small $\boldsymbol{q}_{(1)}$ that $\left\| \boldsymbol{q}_{(1)} \right\| = 1$). We observe for both Option A and Option B that the curves with markers "$\triangle$" overlap the curves with markers "+" at the beginning and the latter part of the overparameterized regime. This phenomenon validates the implication (4) which is inferred from the factor $\min\{r, 1 - r\}$ in Eqs. (11) and (12).

## 3.2 Insights for Option A

**(A1) Benign overfitting w.r.t. $p_{(2)}$ is easier to observe with small knowledge transfer.** In the underparameterized regime, by Eq. (15), $\mathbb{E}[\mathcal{L}]$ is at least $\mathcal{L}_{\text{co}} + \frac{\mathcal{L}_{\text{co}} + \sigma_{(2)}^2}{n_{(2)}}$. In contrast, for the overparameterized regime, when $\mathcal{L}_{\text{co}}$ is large, Term A1 of Eq. (14) dominates $\mathbb{E}[\mathcal{L}]$. When $p_{(2)}$ increases to $\infty$, Term A1 will decrease to $\mathcal{L}_{\text{co}}$. Notice that large $\mathcal{L}_{\text{co}}$ implies small knowledge transfer from the source task to the target task. Thus, *benign overfitting w.r.t. $p_{(2)}$ appears when knowledge transfer is small.*

---

[5]i.e., the descent of the test error stops at a certain point (which is like a floor)

In Fig. 1(b), we let the ground-truth parameters be very small compared with the noise level, so the error $\mathcal{L}$ in Fig. 1 is mainly from noise. The blue curve with markers "$\times$" has larger $\sigma_{(1)}$ (with $\sigma_{(1)} = 3$) compared with the blue curve with markers "$\triangledown$" (with $\sigma_{(1)} = 0.1$), and consequently, larger $\mathcal{L}_{\text{co}}$ and smaller knowledge transfer. We observe from Fig. 1(b) that the blue curve with markers "$\times$" descends w.r.t. $p_{(2)}$ in the entire overparameterized regime, while the blue curve with markers "$\triangledown$" descends at the beginning of the overparameterized regime and ascends in the remainder of the overparameterized regime. Such a phenomenon validates the insight (A1).

**(A2) Larger $p$ is not always good to reduce the noise effect when overparameterized.** By Theorems 1 and 2, we know that the direct effect of $p$ on noise in the overparameterized regime is only through the term $b_{\text{noise}}$ in $\mathcal{L}_{\text{co}}$. By checking the sign of $\frac{\partial b_{\text{noise}}}{\partial p}$, we can prove that $b_{\text{noise}}$ increases w.r.t. $p$ when $p^2 < p_{(1)}(p_{(1)} - n_{(1)} - 1)$, and decreases when $p^2 > p_{(1)}(p_{(1)} - n_{(1)} - 1)$ (see calculation details in Lemma 11 in Appendix A.1).

In Fig. 1(c), the blue curve with markers "$\triangleright$" depicts how the model error $\mathcal{L}$ of Option A changes w.r.t. $p$ in the overparameterized regime ($p + p_{(1)} > n_{(1)}$). This curve first increases and then decreases, which validates the insight (A2).

### 3.3 INSIGHTS FOR OPTION B

**(B1) Benign overfitting w.r.t. $p_{(2)}$ is easier to observe with large knowledge transfer and small target task-specific parameters.** In Eq. (16), small $\mathcal{L}_{\text{co}} + \left\| q_{(2)} \right\|^2$ implies that Term B2 dominates the value of $\mathbb{E}[\mathcal{L}]$. As we explained previously in (2) of Section 3.1, benign overfitting exists in this situation. Meanwhile, small $\mathcal{L}_{\text{co}}$ and $\left\| q_{(2)} \right\|$ imply large knowledge transfer and small target task-specific parameters, respectively.

In Fig. 1(b), the orange curve with markers "$\triangleleft$" denotes the model error $\mathcal{L}$ of Option B w.r.t. $p_{(2)}$ when $\sigma_{(1)}$ and $q_{(2)}$ are small, i.e., large knowledge transfer and small target task-specific parameters. Compared with the orange curve with markers "$\times$", this curve descends in the entire overparameterized regime and can achieve a lower value than that of the underparameterized regime. This phenomenon validates the insight (B1).

**(B2) Multiple descents of noise effect when increasing $p$ in the overparameterized regime.** Different from Option A where $p$ only affects the consequence of the noise in the source task (since no $p$ appears in Eq. (14) except $\mathcal{L}_{\text{co}}$), for Eq. (16) of Option B, we see that $p$ not only affects $\mathcal{L}_{\text{co}}$ but also Term B2, which implies that $p$ relates to the noise effect in both the source task and the target task. Specifically, the trend of $\mathbb{E}[\mathcal{L}]$ w.r.t. $p$ is determined by $(1 - \frac{n_{(2)}}{p + p_{(2)}})b_{\text{noise}}$ and Term B2 in Eq. (16). In (A2) of Section 3.2, we show that $b_{\text{noise}}$ sometimes first increases and then decreases. The factor $1 - \frac{n_{(2)}}{p + p_{(2)}}$ is monotone increasing w.r.t. $p$. Term B2 in Eq. (16) is monotone decreasing w.r.t. $p$. Thus, the overall noise effect may have multiple descents w.r.t. $p$.

In Fig. 1(c), the orange curve with markers "$\triangleright$" provides an example of how the model error $\mathcal{L}$ of Option B behaves in the overparameterized regime. We see that this curve has multiple descents, which validates the insight (B2). We also run additional simulations in Appendix F with a neural network, and we can observe the descent w.r.t. the number of parameters of the transferred part.

## 4 FURTHER DISCUSSION

### 4.1 WHICH OPTION PERFORMS BETTER IN THE OVERPARAMETERIZED REGIME?

**(C1)** First, by comparing the coefficients of $\mathcal{L}_{\text{co}}$ in Eq. (14) and Eq. (16), we know that the effect of the error in step one deteriorates in the model error $\mathcal{L}$ of Option A (since the coefficient of $\mathcal{L}_{\text{co}}$ in Eq. (14) is larger than 1), whereas this is mitigated in the model error of Option B (since the coefficient of $\mathcal{L}_{\text{co}}$ in Eq. (16) is smaller than 1). **(C2)** Second, by comparing the coefficients of $\left\| q_{(2)} \right\|^2$ and $\sigma_{(2)}^2$ in Eqs. (14) and (16) under the same $p$ and $p_{(2)}$, we know that Option B is worse to learn $q_{(2)}$ but is better to reduce the noise effect of $\sigma_{(2)}$ than Option A (since $1 - \frac{n_{(2)}}{p_{(2)}} < 1 - \frac{n_{(2)}}{p + p_{(2)}}$

and $\frac{n_{(2)}}{p_{(2)}-n_{(2)}-1} > \frac{n_{(2)}}{p+p_{(2)}-n_{(2)}-1}$). **(C3)** Third, by letting $p_{(2)} \to \infty$ in Eqs. (14) and (16), the model error $\mathcal{L}$ of both Option A and Option B approaches the same value $\mathcal{L}_{\text{co}} + \left\|\boldsymbol{q}_{(2)}\right\|^2$.

**Intuitive Comparison of Options A and B:** An intuitive explanation of the reason for these differences is that Option B does train the common part learned by the source task but Option A does not. Thus, Option B should do better to learn the common part. At the same time, since Option B uses more parameters ($p + p_{(2)}$) than Option A ($p$) to learn the target task's samples, the noise effect is spread among more parameters in Option B than in Option A, and thus Option B can mitigate the noise better than Option A. However, those additional $p$ parameters interfere with the learning of $\boldsymbol{q}_{(2)}$ since those $p$ parameters correspond to the features of the common part $\hat{\mathcal{S}}_{\text{co}}$, not the target task-specific features $\hat{\mathcal{S}}_{(2)}$, which implies that Option B is worse in learning $\boldsymbol{q}_{(2)}$ than Option A.

In Fig. 1(b), when overparameterized (i.e., $p_{(2)} > 50$ for Option A, and $p_{(2)} > 30$ for Option B), Option A is slightly better than Option B around $p_{(2)} = 70$ under the situation "$\sigma_{(1)} = 0.1$, $\sigma_{(2)} = 0.2$, $\left\|\boldsymbol{q}_{(2)}\right\| = 1$" (i.e., the two curves with markers "+"). Notice that this situation has the smallest $\sigma_{(1)}, \sigma_{(2)}$ and the largest $\left\|\boldsymbol{q}_{(2)}\right\|$. Thus, insights (C1),(C2) are verified. Besides, in Fig. 1(b), in every situation, the curves of Option A and Option B overlap when $p_{(2)}$ is very large, which validates insight (C3).

### 4.2 THE COMMON PART OR THE TASK-SPECIFIC PART?

**When the total number of parameters is fixed, it is better to use more parameters on the task-specific parts.** Specifically, we have the following proposition.

**Proposition 4.** *When $p + p_{(1)} = C$ is fixed, $\mathcal{L}_{co}$ is monotone increasing with respect to $p$. Therefore, in order to minimize $\mathcal{L}_{co}$ when Definition 1 is assured, the best choice is $p = s$, $p_{(1)} = C - s$.*

**Sometimes it is even better to sacrifice certain *true* features in the common part in favor of employing more *redundant* features in the task-specific part.** We still consider the case of fixed $p + p_{(1)} = C$. In certain situations (especially when the noise level is large and some true parameters are very small), it is better to make $p$ even smaller than $s$, i.e., it is better to violate Definition 1 deliberately (in contrast to Remark 1 where Definition 1 is violated unconsciously). We now construct an example of this situation. Let $\left\|\boldsymbol{q}_{(1)}\right\|^2 = 0$, $\left\|\boldsymbol{w}_{(2)}\right\| + \left\|\boldsymbol{w}_{(1)}\right\| = 1$ (so $\bar{b}_2^2 \leq 1$ by Eq. (11)). Suppose there are only 2 true common features (i.e., $s = 2$) and $C > n_{(1)} + 1$. If we do not violate Definition 1, then by Proposition 4, the best choice is to let $p = 2$. By Theorem 1 we know that $\mathcal{L}_{\text{co}}$ is at least $\mathcal{Q}_1 := \frac{2}{C} \cdot \frac{n_{(1)}\sigma_{(1)}^2}{C-n_{(1)}-1}$ (since $\mathcal{L}_{\text{co}}^{\text{noiseless}} \geq 0$). In contrast, if we violate Definition 1 deliberately by sacrificing one true common feature with parameter value 0.1 for the source task and value 0 for the target task, then the only effect is enlarging the source task's noise level by $\sigma_{(1)}^2 \leftarrow \sigma_{(1)}^2 + 0.1^2$. Thus, by Theorem 1, we know that $\mathcal{L}_{\text{co}}$ is at most $\mathcal{Q}_2 := 1 + \frac{1}{C} \cdot \frac{n_{(1)}(\sigma_{(1)}^2 + 0.1^2)}{C-n_{(1)}-1}$ (since $\bar{b}_2^2 \leq 1$). We can easily find a large enough $\sigma_{(1)}^2$ to make $\mathcal{Q}_1 > \mathcal{Q}_2$, which leads to our conclusion.

## 5 CONCLUSION

Our study on transfer learning in linear regression models provides valuable insights into the generalization performance of the target task. We propose a comprehensive framework that considers task similarity in terms of both parameter distance and feature sets. Our analysis characterizes the double descent of transfer learning for two different options of parameter transfer. Further investigation reveals that allocating more redundant features to the task-specific part, rather than the common part, can enhance performance when the total number of features is fixed. Moreover, sometimes sacrificing true features in the common part in favor of employing more redundant features in the task-specific part can yield notable benefits, especially in scenarios with high noise levels and small numbers of true parameters. These findings contribute to a better understanding of transfer learning and offer practical guidance for designing effective transfer learning approaches.

There are some interesting directions for future work. First, we can use our current framework of partial similarity to analyze the performance of sample transfer. Second, going beyond the linear models of Gaussian features, we can use models that are closer to actual DNNs (such as neural tangent kernel models) to study the generalization performance of overfitted transfer learning.

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

# Supplemental Material

## A   SOME USEFUL LEMMAS

We first introduce some useful lemmas that will be used in the proofs of our main results.

**Lemma 5.** *Consider $n$ training samples and $p$ features. The stacked training input is $\mathbf{X} \in \mathbb{R}^{p \times n}$ and the corresponding output is $\boldsymbol{y} \in \mathbb{R}^n$. If GD/SGD on linear regression with mean-square-error converges to zero (i.e., overfitted), then the convergence point is the solution that has the minimum $\ell_2$-norm of the change of parameters from the initial point.*

*Proof.* Let $\boldsymbol{a}$ denote the parameters to train and $\boldsymbol{a}_0$ the initial point. Since GD/SGD is used in a linear model, then in each iteration, the change of the parameters must be in the column space of $\mathbf{X}$. To realize this, we can simply calculate the gradient for the $i$-th training sample as

$$\frac{\partial (y_i - \boldsymbol{x}_i^T \boldsymbol{a})^2}{\partial \boldsymbol{a}} = -2(y_i - \boldsymbol{x}_i^T \boldsymbol{a})\boldsymbol{x}_i,$$

where $\boldsymbol{x}_i$ is the $i$-th column of $\mathbf{X}$ and $y_i$ is the $i$-th element of $\boldsymbol{y}$. As we can see, the gradient is always parallel to one column of $\mathbf{X}$, which implies that the overall change of the parameters is still in the column space of $\mathbf{X}$. Thus, we can always find $\boldsymbol{b} \in \mathbb{R}^p$ such that

$$\boldsymbol{a} - \boldsymbol{a}_0 = \mathbf{X}\boldsymbol{b}. \tag{18}$$

For the convergence point that makes the training loss become zero, we have

$$\mathbf{X}^T \boldsymbol{a} = \boldsymbol{y}. \tag{19}$$

Substituting Eq. (18) into Eq. (19), we have

$$\mathbf{X}^T (\boldsymbol{a}_0 + \mathbf{X}\boldsymbol{b}) = \boldsymbol{y}$$
$$\implies \boldsymbol{b} = (\mathbf{X}^T \mathbf{X})^{-1} (\boldsymbol{y} - \mathbf{X}^T \boldsymbol{a}_0)$$
$$\implies \boldsymbol{a} = \boldsymbol{a}_0 + \mathbf{X}(\mathbf{X}^T \mathbf{X})^{-1} (\boldsymbol{y} - \mathbf{X}^T \boldsymbol{a}_0),$$

which is exactly the minimum $\ell_2$-norm solution of the following linear problem:

$$\min_{\boldsymbol{a}} \quad \|\boldsymbol{a} - \boldsymbol{a}_0\|$$
$$\text{subject to} \quad \mathbf{X}^T (\boldsymbol{a} - \boldsymbol{a}_0) = \boldsymbol{y} - \mathbf{X}^T \boldsymbol{a}_0.$$

The constraint is also equivalent to $\mathbf{X}_{(1)}^T \boldsymbol{a} = \boldsymbol{y}$. The result of this lemma thus follows. □

**Lemma 6** (Cauchy-Schwarz inequality on random vectors)**.** *Consider any two random vectors $\boldsymbol{a}, \boldsymbol{b} \in \mathbb{R}^d$. We must have*

$$\left| \mathbb{E}[\boldsymbol{a}^T \boldsymbol{b}] \right| \leq \sqrt{\mathbb{E}[\|\boldsymbol{a}\|^2] \cdot \mathbb{E}[\|\boldsymbol{b}\|^2]}. \tag{20}$$

*Consequently, we have*

$$\mathbb{E}[\|\boldsymbol{a} + \boldsymbol{b}\|^2] \leq \left( \sqrt{\mathbb{E}[\|\boldsymbol{a}\|^2]} + \sqrt{\mathbb{E}[\|\boldsymbol{b}\|^2]} \right)^2, \tag{21}$$

*and*

$$\mathbb{E}[\|\boldsymbol{a} + \boldsymbol{b}\|^2] \geq \left( \sqrt{\mathbb{E}[\|\boldsymbol{a}\|^2]} - \sqrt{\mathbb{E}[\|\boldsymbol{b}\|^2]} \right)^2. \tag{22}$$

*Proof.* Eq. (20) directly follows by Cauchy-Schwarz inequality after defining the inner product on the set of random vectors using the expectation of their inner product:

$$\langle \boldsymbol{a}, \, \boldsymbol{b} \rangle_\mu := \int \boldsymbol{a}^T \boldsymbol{b} \, d\mu = \mathbb{E}[\boldsymbol{a}^T \boldsymbol{b}],$$

where $\mu(\cdot)$ denotes the probability measure. We can easily verify the following properties of $\langle \cdot, \cdot \rangle_\mu$:

$$\langle \boldsymbol{a}, \ \boldsymbol{b} \rangle_\mu = \langle \boldsymbol{b}, \ \boldsymbol{a} \rangle_\mu, \text{ (conjugate symmetry)};$$
$$\langle \lambda_1 \boldsymbol{a} + \lambda_2 \boldsymbol{b}, \ \boldsymbol{c} \rangle_\mu = \lambda_1 \langle \boldsymbol{a}, \ \boldsymbol{c} \rangle_\mu + \lambda_2 \langle \boldsymbol{b}, \ \boldsymbol{c} \rangle_\mu, \text{ (linearity)};$$
$$\langle \boldsymbol{a}, \ \boldsymbol{a} \rangle_\mu > 0 \text{ for all } \boldsymbol{a} \neq \boldsymbol{0}. \text{ (positive-definiteness)}.$$

Thus, $\langle \cdot, \cdot \rangle_\mu$ defines a valid inner product space. It remains to prove Eq. (21) and Eq. (22). To that end, we have

$$
\begin{aligned}
\mathbb{E}[\|\boldsymbol{a} + \boldsymbol{b}\|^2] &= \mathbb{E}[\|\boldsymbol{a}\|^2] + \mathbb{E}[\boldsymbol{a}^T \boldsymbol{b}] + 2\,\mathbb{E}[\|\boldsymbol{b}\|^2] \\
&\leq \mathbb{E}[\|\boldsymbol{a}\|^2] + 2\left|\mathbb{E}[\boldsymbol{a}^T \boldsymbol{b}]\right| + \mathbb{E}[\|\boldsymbol{b}\|^2] \\
&\leq \mathbb{E}[\|\boldsymbol{a}\|^2] + 2\sqrt{\mathbb{E}[\|\boldsymbol{a}\|^2] \cdot \mathbb{E}[\|\boldsymbol{b}\|^2]} + \mathbb{E}[\|\boldsymbol{b}\|^2] \text{ (by Eq. (20))} \\
&= \left(\sqrt{\mathbb{E}[\|\boldsymbol{a}\|^2]} + \sqrt{\mathbb{E}[\|\boldsymbol{b}\|^2]}\right)^2.
\end{aligned}
$$

Similarly, we have

$$
\begin{aligned}
\mathbb{E}[\|\boldsymbol{a} + \boldsymbol{b}\|^2] &= \mathbb{E}[\|\boldsymbol{a}\|^2] + \mathbb{E}[\boldsymbol{a}^T \boldsymbol{b}] + 2\,\mathbb{E}[\|\boldsymbol{b}\|^2] \\
&\geq \mathbb{E}[\|\boldsymbol{a}\|^2] - 2\left|\mathbb{E}[\boldsymbol{a}^T \boldsymbol{b}]\right| + \mathbb{E}[\|\boldsymbol{b}\|^2] \\
&\geq \mathbb{E}[\|\boldsymbol{a}\|^2] - 2\sqrt{\mathbb{E}[\|\boldsymbol{a}\|^2] \cdot \mathbb{E}[\|\boldsymbol{b}\|^2]} + \mathbb{E}[\|\boldsymbol{b}\|^2] \text{ (by Eq. (20))} \\
&= \left(\sqrt{\mathbb{E}[\|\boldsymbol{a}\|^2]} - \sqrt{\mathbb{E}[\|\boldsymbol{b}\|^2]}\right)^2.
\end{aligned}
$$

The result of this lemma therefore follows. $\qquad\square$

The result of the following lemma can be found in the literature (e.g., (Belkin et al., 2020)).

**Lemma 7.** *Consider a random matrix* $\mathbf{K} \in \mathbb{R}^{p \times n}$ *where $p$ and $n$ are two positive integers and $p > n + 1$. Each element of $\mathbf{K}$ is* i.i.d. *according to standard Gaussian distribution. For any fixed vector $\boldsymbol{a} \in \mathbb{R}^p$, we must have*

$$\mathbb{E}\left\|\left(\mathbf{I}_p - \mathbf{K}\left(\mathbf{K}^T\mathbf{K}\right)^{-1}\mathbf{K}^T\right)\boldsymbol{a}\right\|^2 = \left(1 - \frac{n}{p}\right)\|\boldsymbol{a}\|^2,$$

$$\mathbb{E}\left\|\mathbf{K}\left(\mathbf{K}^T\mathbf{K}\right)^{-1}\mathbf{K}^T\boldsymbol{a}\right\|^2 = \frac{n}{p}\|\boldsymbol{a}\|^2.$$

*Further, when $p \geq 16$, we have*

$$\Pr\left\{\left\|\left(\mathbf{I}_p - \mathbf{K}\left(\mathbf{K}^T\mathbf{K}\right)^{-1}\mathbf{K}^T\right)\boldsymbol{a}\right\|^2 \leq \frac{p - n + 2\sqrt{(p-n)\ln p} + 2\ln p}{p - 2\sqrt{p\ln p}}\|\boldsymbol{a}\|^2\right\} \geq 1 - \frac{2}{p}.$$

$$\Pr\left\{\left\|\mathbf{K}\left(\mathbf{K}^T\mathbf{K}\right)^{-1}\mathbf{K}^T\boldsymbol{a}\right\|^2 \leq \frac{n + 2\sqrt{n\ln n} + 2\ln n}{p - 2\sqrt{p\ln p}}\|\boldsymbol{a}\|^2\right\} \geq 1 - \frac{1}{p} - \frac{1}{n}.$$

*Proof.* Define $\mathbf{P}_0 := \mathbf{K}\left(\mathbf{K}^T\mathbf{K}\right)^{-1}\mathbf{K}^T$. By the definition of $\mathbf{P}_0$, we have $\mathbf{P}_0\mathbf{P}_0 = \mathbf{P}_0$. Thus, $\mathbf{P}_0$ is a projection that projects a $p$-dim vector to a $n$-dim subspace (i.e., the column space of $\mathbf{K}$). Since each element of $\mathbf{K}$ is *i.i.d.* following standard Gaussian distribution, we can conclude that $\mathbf{P}_0$ has rotational symmetry. The rest of the calculation and proof is similar to that of Proposition 3 of (Ju et al., 2023). Here we give an intuitive explanation of why $\mathbb{E}\left\|(\mathbf{I}_p - \mathbf{P}_0)\boldsymbol{a}\right\|^2 = \left(1 - \frac{n}{p}\right)\|\boldsymbol{a}\|^2$. The rotational symmetry implies that the projection on $\boldsymbol{a}$ makes the change of the squared norm proportional to the dimension of the original space and the subspace, i.e., $E\left\|\mathbf{P}_0\boldsymbol{a}\right\|^2 = \frac{n}{p}\|\boldsymbol{a}\|^2$ and consequently, $\mathbb{E}\left\|\boldsymbol{a} - \mathbf{P}_0\boldsymbol{a}\right\|^2 = \left(1 - \frac{n}{p}\right)\|\boldsymbol{a}\|^2$. $\qquad\square$

**Lemma 8.** *Consider a random matrix* $\mathbf{K} \in \mathbb{R}^{a \times b}$ *where* $a > b + 1$. *Each element of* $\mathbf{K}$ *is i.i.d. following standard Gaussian distribution* $\mathcal{N}(0, 1)$. *Consider three Gaussian random vectors* $\boldsymbol{\alpha}, \boldsymbol{\gamma} \in \mathbf{R}^a$ *and* $\boldsymbol{\beta} \in \mathbf{R}^b$ *such that* $\boldsymbol{\alpha} \sim \mathcal{N}(\mathbf{0}, \sigma_\alpha^2 \mathbf{I}_a)$, $\boldsymbol{\gamma} \sim \mathcal{N}(\mathbf{0}, \mathrm{diag}(d_1^2, d_2^2, \cdots, d_a^2))$, *and* $\boldsymbol{\beta} \sim \mathcal{N}(\mathbf{0}, \sigma_\beta^2 \mathbf{I}_b)$. *Here* $\mathbf{K}$, $\boldsymbol{\alpha}$, $\boldsymbol{\gamma}$, *and* $\boldsymbol{\beta}$ *are independent of each other. We then must have*

$$\mathbb{E}\left[(\mathbf{K}^T\mathbf{K})^{-1}\right] = \frac{\mathbf{I}_b}{a - b - 1}, \tag{23}$$

$$\mathbb{E}\left\|\mathbf{K}(\mathbf{K}^T\mathbf{K})^{-1}\boldsymbol{\beta}\right\|^2 = \frac{b\sigma_\beta^2}{a - b - 1}, \tag{24}$$

$$\mathbb{E}\left\|(\mathbf{K}^T\mathbf{K})^{-1}\mathbf{K}^T\boldsymbol{\alpha}\right\|^2 = \frac{b\sigma_\alpha^2}{a - b - 1}, \tag{25}$$

$$\mathbb{E}\left\|(\mathbf{K}^T\mathbf{K})^{-1}\mathbf{K}^T\boldsymbol{\gamma}\right\|^2 = \frac{b\sum_{i=1}^a d_i^2}{a(a - b - 1)}. \tag{26}$$

*Proof.* By the definition of $\mathbf{K}$, we know that $(\mathbf{K}^T\mathbf{K})^{-1}$ follows inverse-Wishart distribution, and thus Eq. (23) follows. (The expression and derivation of the mean of any inverse-Wishart matrix can be found in the literature, e.g., (Mardia, 1979).) It remains to prove Eq. (24) and Eq. (25). To that end, we obtain

$$\begin{aligned}
\mathbb{E}\left\|\mathbf{K}(\mathbf{K}^T\mathbf{K})^{-1}\boldsymbol{\beta}\right\|^2 &= \mathbb{E}\left[\left(\mathbf{K}(\mathbf{K}^T\mathbf{K})^{-1}\boldsymbol{\beta}\right)^T \left(\mathbf{K}(\mathbf{K}^T\mathbf{K})^{-1}\boldsymbol{\beta}\right)\right] \\
&= \mathbb{E}\left[\boldsymbol{\beta}^T(\mathbf{K}^T\mathbf{K})^{-1}\boldsymbol{\beta}\right] \\
&= \mathbb{E}_{\boldsymbol{\beta}}\left[\boldsymbol{\beta}^T \mathbb{E}_{\mathbf{K}}[(\mathbf{K}^T\mathbf{K})^{-1}]\boldsymbol{\beta}\right] \text{ (since } \mathbf{K} \text{ and } \boldsymbol{\beta} \text{ are independent)} \\
&= \frac{1}{a - b - 1}\mathbb{E}\left[\boldsymbol{\beta}^T\boldsymbol{\beta}\right] \text{ (by Eq. (23))} \\
&= \frac{b\sigma_\beta^2}{a - b - 1} \text{ (since } \mathbb{E}_{\boldsymbol{\beta}}[\boldsymbol{\beta}^T\boldsymbol{\beta}] = b\sigma_\beta^2 \text{ because } \boldsymbol{\beta} \sim \mathcal{N}(\mathbf{0}, \sigma_\beta^2 \mathbf{I}_b)),
\end{aligned}$$

and

$$\begin{aligned}
&\mathbb{E}\left\|(\mathbf{K}^T\mathbf{K})^{-1}\mathbf{K}^T\boldsymbol{\alpha}\right\|^2 \\
&= \mathbb{E}\left[\left((\mathbf{K}^T\mathbf{K})^{-1}\mathbf{K}^T\boldsymbol{\alpha}\right)^T \left((\mathbf{K}^T\mathbf{K})^{-1}\mathbf{K}^T\boldsymbol{\alpha}\right)\right] \\
&= \mathbb{E}\left[\boldsymbol{\alpha}^T\mathbf{K}(\mathbf{K}^T\mathbf{K})^{-1}(\mathbf{K}^T\mathbf{K})^{-1}\mathbf{K}^T\boldsymbol{\alpha}\right] \\
&= \mathbb{E}\left[\mathrm{Tr}\left(\boldsymbol{\alpha}^T\mathbf{K}(\mathbf{K}^T\mathbf{K})^{-1}(\mathbf{K}^T\mathbf{K})^{-1}\mathbf{K}^T\boldsymbol{\alpha}\right)\right] \\
&= \mathbb{E}\left[\mathrm{Tr}\left(\mathbf{K}(\mathbf{K}^T\mathbf{K})^{-1}(\mathbf{K}^T\mathbf{K})^{-1}\mathbf{K}^T\boldsymbol{\alpha}\boldsymbol{\alpha}^T\right)\right] \text{ (by trace trick that } \mathrm{Tr}(\mathbf{AB}) = \mathrm{Tr}(\mathbf{BA})) \\
&= \mathbb{E}_{\mathbf{K}}\left[\mathrm{Tr}\left(\mathbf{K}(\mathbf{K}^T\mathbf{K})^{-1}(\mathbf{K}^T\mathbf{K})^{-1}\mathbf{K}^T \mathbb{E}_{\boldsymbol{\alpha}}[\boldsymbol{\alpha}\boldsymbol{\alpha}^T]\right)\right] \text{ (since } \mathbf{K} \text{ and } \boldsymbol{\alpha} \text{ are independent)} \qquad (27) \\
&= \sigma_\alpha^2 \mathbb{E}\left[\mathrm{Tr}\left(\mathbf{K}(\mathbf{K}^T\mathbf{K})^{-1}(\mathbf{K}^T\mathbf{K})^{-1}\mathbf{K}^T\right)\right] \text{ (since } \mathbb{E}_{\boldsymbol{\alpha}}[\boldsymbol{\alpha}\boldsymbol{\alpha}^T] = \sigma_\alpha^2 \mathbf{I}_a \text{ because } \boldsymbol{\alpha} \sim \mathcal{N}(\mathbf{0}, \sigma_\alpha^2 \mathbf{I}_a)) \\
&= \sigma_\alpha^2 \mathbb{E}\left[\mathrm{Tr}\left((\mathbf{K}^T\mathbf{K})^{-1}(\mathbf{K}^T\mathbf{K})^{-1}\mathbf{K}^T\mathbf{K}\right)\right] \text{ (by trace trick that } \mathrm{Tr}(\mathbf{AB}) = \mathrm{Tr}(\mathbf{BA})) \\
&= \sigma_\alpha^2 \mathrm{Tr}\left(\mathbb{E}[(\mathbf{K}^T\mathbf{K})^{-1}]\right) \\
&= \frac{b\sigma_\alpha^2}{a - b - 1} \text{ (by Eq. (23))}.
\end{aligned}$$

Thus, we have proven Eq. (24) and Eq. (25). It remains to prove Eq. (26). To that end, we define $\boldsymbol{e}_i$ as the $i$-th standard basis. Define $\mathbf{M}_j \in \mathbb{R}^{a \times a}$ as the permutation matrix that switches the first element with the $j$-th element for $j = 2, 3, \cdots, a$. Since $\mathbf{M}_j$ is a permutation matrix, we have $\mathbf{M}_j^T\mathbf{M}_j = \mathbf{I}_a$. Thus, for all $j = 2, 3, \cdots, a$, we obtain

$$(\mathbf{K}^T\mathbf{K})^{-1}\mathbf{K}^T\boldsymbol{e}_j = (\mathbf{K}^T\mathbf{M}_j^T\mathbf{M}_j\mathbf{K})^{-1}\mathbf{K}^T\mathbf{M}_j^T\mathbf{M}_j\boldsymbol{e}_j = (\hat{\mathbf{K}}^T\hat{\mathbf{K}})^{-1}\hat{\mathbf{K}}^T\boldsymbol{e}_1,$$

where $\hat{\mathbf{K}} := \mathbf{M}_j\mathbf{K}$. By definition of $\mathbf{K}$, we know that $\hat{\mathbf{K}}$ and $\mathbf{K}$ have the same distribution. Thus, we have

$$\mathbb{E}\left\|(\mathbf{K}^T\mathbf{K})^{-1}\mathbf{K}^T\boldsymbol{e}_j\right\|^2 = \mathbb{E}\left\|(\mathbf{K}^T\mathbf{K})^{-1}\mathbf{K}^T\boldsymbol{e}_1\right\|^2 \text{ for all } j = 2, 3, \cdots, b. \tag{28}$$

We then derive

$$
\begin{aligned}
&\mathbb{E}\left\|(\mathbf{K}^T\mathbf{K})^{-1}\mathbf{K}^T\boldsymbol{\gamma}\right\|^2 \\
=&\underset{\mathbf{K}}{\mathbb{E}}\left[\operatorname{Tr}\left(\mathbf{K}(\mathbf{K}^T\mathbf{K})^{-1}(\mathbf{K}^T\mathbf{K})^{-1}\mathbf{K}^T\underset{\boldsymbol{\gamma}}{\mathbb{E}}[\boldsymbol{\gamma}\boldsymbol{\gamma}^T]\right)\right] \text{ (similar to Eq. (27))} \\
=&\mathbb{E}\left[\operatorname{Tr}\left(\mathbf{K}(\mathbf{K}^T\mathbf{K})^{-1}(\mathbf{K}^T\mathbf{K})^{-1}\mathbf{K}\operatorname{diag}(d_1^2,\cdots,d_a^2)\right)\right] \text{ (by the definition of } \boldsymbol{\gamma}) \\
=&\sum_{i=1}^{a}\mathbb{E}\left[\operatorname{Tr}\left(\mathbf{K}(\mathbf{K}^T\mathbf{K})^{-1}(\mathbf{K}^T\mathbf{K})^{-1}\mathbf{K}\operatorname{diag}(0,\cdots,0,\underbrace{d_i^2}_{i\text{-th element}},0,\cdots,0)\right)\right] \\
=&\sum_{i=1}^{a}d_i^2\,\mathbb{E}\left\|(\mathbf{K}^T\mathbf{K})^{-1}\mathbf{K}^T\boldsymbol{e}_i\right\|^2 \text{ (similar to Eq. (27))} \\
=&\frac{\sum_{j=1}^{a}d_j^2}{a}\sum_{i=1}^{a}\mathbb{E}\left\|(\mathbf{K}^T\mathbf{K})^{-1}\mathbf{K}^T\boldsymbol{e}_i\right\|^2 \text{ (by Eq. (28))} \\
=&\frac{\sum_{j=1}^{a}d_j^2}{a}\sum_{i=1}^{a}\mathbb{E}\left[\operatorname{Tr}\left(\mathbf{K}(\mathbf{K}^T\mathbf{K})^{-1}(\mathbf{K}^T\mathbf{K})^{-1}\mathbf{K}\operatorname{diag}(0,\cdots,0,\underbrace{1}_{i\text{-th element}},0,\cdots,0)\right)\right] \\
&\text{(similar to Eq. (27))} \\
=&\frac{\sum_{j=1}^{a}d_j^2}{a}\,\mathbb{E}\left[\operatorname{Tr}\left(\mathbf{K}(\mathbf{K}^T\mathbf{K})^{-1}(\mathbf{K}^T\mathbf{K})^{-1}\mathbf{K}\right)\right] \\
=&\frac{\sum_{j=1}^{a}d_j^2}{a}\,\mathbb{E}\left[\operatorname{Tr}\left((\mathbf{K}^T\mathbf{K})^{-1}(\mathbf{K}^T\mathbf{K})^{-1}\mathbf{K}^T\mathbf{K}\right)\right] \text{ (by trace trick that } \operatorname{Tr}(\mathbf{AB})=\operatorname{Tr}(\mathbf{BA})) \\
=&\frac{\sum_{j=1}^{a}d_j^2}{a}\operatorname{Tr}\left(\mathbb{E}[(\mathbf{K}^T\mathbf{K})^{-1}]\right) \\
=&\frac{b\sum_{j=1}^{a}d_j^2}{a(a-b-1)} \text{ (by Eq. (23)).}
\end{aligned}
$$

We have therefore proven Eq. (26), and the result of this lemma follows. $\qquad\square$

## A.1 CALCULATION OF DERIVATIVE

**Lemma 9.** *Consider the expression of $\mathbb{E}[\mathcal{L}]$ in Eq. (14) (overparameterized regime of Option A). When $\mathcal{L}_{co}+\sigma_{(2)}^2 \geq \left\|\boldsymbol{q}_{(2)}\right\|^2$, $\mathbb{E}[\mathcal{L}]$ is monotone decreasing w.r.t. $p_{(2)}$. When $\mathcal{L}_{co}+\sigma_{(2)}^2 < \left\|\boldsymbol{q}_{(2)}\right\|^2$, we have*

$$
\frac{\partial\,\mathbb{E}[\mathcal{L}]}{\partial p_{(2)}}
\begin{cases}
\leq 0, \text{ if } p_{(2)} \in \left(n_{(2)}+1,\ \dfrac{n_{(2)}+1}{1-\frac{\sqrt{\mathcal{L}_{co}+\sigma_{(2)}^2}}{\left\|\boldsymbol{q}_{(2)}\right\|}}\right], \\[3ex]
> 0, \text{ if } p_{(2)} \in \left(\dfrac{n_{(2)}+1}{1-\frac{\sqrt{\mathcal{L}_{co}+\sigma_{(2)}^2}}{\left\|\boldsymbol{q}_{(2)}\right\|}},\ \infty\right).
\end{cases}
$$

*Proof.* We obtain

$$
\begin{aligned}
\frac{\partial\,\mathbb{E}[\mathcal{L}]}{\partial p_{(2)}} &= -\frac{n_{(2)}\left(\mathcal{L}_{co}+\sigma_{(2)}^2\right)}{(p_{(2)}-n_{(2)}-1)^2}+\frac{n_{(2)}}{p_{(2)}^2}\left\|\boldsymbol{q}_{(2)}\right\|^2 \\
&= \frac{n_{(2)}\left\|\boldsymbol{q}_{(2)}\right\|^2}{(p_{(2)}-n_{(2)}-1)^2}\left(-\frac{\mathcal{L}_{co}+\sigma_{(2)}^2}{\left\|\boldsymbol{q}_{(2)}\right\|^2}+\left(1-\frac{n_{(2)}+1}{p_{(2)}}\right)^2\right).
\end{aligned}
$$

The result of this lemma thus follows by checking the sign of the above expression. $\qquad\square$

**Lemma 10.** *Define*

$$t := \frac{n_{(2)} + 1}{1 - \frac{\sigma_{(2)}}{\sqrt{\mathcal{L}_{co} + \|\boldsymbol{q}_{(2)}\|^2}}}.$$

*Consider the expression of $\mathbb{E}[\mathcal{L}]$ in Eq. (16). When $\sigma_{(2)}^2 \geq \mathcal{L}_{co} + \|\boldsymbol{q}_{(2)}\|^2$ or $p \geq t$, $\mathbb{E}[\mathcal{L}]$ is monotone decreasing w.r.t. $p_{(2)}$ in the overparameterized regime. When $\sigma_{(2)}^2 < \mathcal{L}_{co} + \|\boldsymbol{q}_{(2)}\|^2$ and $p < t$, we have*

$$\frac{\partial \mathbb{E}[\mathcal{L}]}{\partial p_{(2)}} \begin{cases} \leq 0, & \text{if } p_{(2)} \in \left(n_{(2)} + 1, \, t - p\right], \\ > 0, & \text{if } p_{(2)} \in (t - p, \, \infty). \end{cases}$$

*Proof.* We obtain

$$\frac{\partial \mathbb{E}[\mathcal{L}]}{\partial p_{(2)}} = \frac{n_{(2)}}{(p + p_{(2)})^2} \left(\mathcal{L}_{co} + \|\boldsymbol{q}_{(2)}\|^2\right) - \frac{n_{(2)}\sigma_{(2)}^2}{(p + p_{(2)} - n_{(2)} - 1)^2},$$

$$= \frac{n_{(2)} \left(\mathcal{L}_{co} + \|\boldsymbol{q}_{(2)}\|^2\right)}{(p + p_{(2)} - n_{(2)} - 1)^2} \left(\left(1 - \frac{n_{(2)} + 1}{p + p_{(2)}}\right)^2 - \frac{\sigma_{(2)}^2}{\mathcal{L}_{co} + \|\boldsymbol{q}_{(2)}\|^2}\right).$$

The result of this lemma thus follows by checking the sign of the above expression. □

**Lemma 11.** *When $p + p_{(1)} > n_{(1)} + 1$, we have*

$$\frac{\partial b_{noise}}{\partial p} \begin{cases} < 0, & \text{when } p^2 > p_{(1)} \left(p_{(1)} - n_{(1)} - 1\right), \\ \geq 0, & \text{otherwise}. \end{cases}$$

*Proof.* By Eq. (13), we obtain

$$\frac{n_{(1)}\sigma_{(1)}^2}{b_{\text{noise}}} = \left(1 + \frac{p_{(1)}}{p}\right) \left(p + p_{(1)} - n_{(1)} - 1\right).$$

In order to determine the sign of $\frac{\partial b_{\text{noise}}}{\partial p}$, it is equivalent to check the sign of $-\frac{\partial(n_{(1)}\sigma_{(1)}^2/b_{\text{noise}})}{\partial p}$. To that end, we have

$$-\frac{\partial(n_{(1)}\sigma_{(1)}^2/b_{\text{noise}})}{\partial p} = \frac{p_{(1)}}{p^2} \left(p + p_{(1)} - n_{(1)} - 1\right) - 1 - \frac{p_{(1)}}{p}$$

$$= \frac{p_{(1)} \left(p_{(1)} - n_{(1)} - 1\right) - p^2}{p^2}.$$

The result of this lemma therefore follows. □

Let $\lambda_{\min}(\cdot)$ and $\lambda_{\min}(\cdot)$ denote the minimum and maximum singular value of a matrix.

**Lemma 12** (Corollary 5.35 of (Vershynin, 2010)). *Let $\mathbf{A}$ be an $N_1 \times N_2$ matrix $(N_1 > N_2)$ whose entries are independent standard normal random variables. Then for every $t \geq 0$, with probability at least $1 - 2\exp(-t^2/2)$, one has*

$$\sqrt{N_1} - \sqrt{N_2} - t \leq \lambda_{\min}(\mathbf{A}) \leq \lambda_{\max}(\mathbf{A}) \leq \sqrt{N_1} + \sqrt{N_2} + t.$$

**Lemma 13** (stated on pp. 1325 of (Laurent & Massart, 2000)). *Let $U$ follow $\chi^2$ distribution with $D$ degrees of freedom. For any positive x, we have*

$$\Pr\left\{U - D \geq 2\sqrt{Dx} + 2x\right\} \leq e^{-x},$$

$$\Pr\left\{D - U \geq 2\sqrt{Dx}\right\} \leq e^{-x}.$$

*By the union bound, we thus obtain*

$$\Pr\left\{U \in \left[D - 2\sqrt{Dx}, \, D + 2\sqrt{Dx} + 2x\right]\right\} \geq 1 - 2e^{-x}.$$

**Lemma 14.** *Let $\mathbf{K} \in \mathbb{R}^{p \times n}$ where $n > p$. We have*

$$\left\|(\mathbf{K}\mathbf{K}^T)^{-1}\mathbf{K}\boldsymbol{a}\right\|^2 \leq \frac{\|\boldsymbol{a}\|^2}{\min \operatorname{Eig}(\mathbf{K}\mathbf{K}^T)} = \frac{\|\boldsymbol{a}\|^2}{\lambda_{\min}^2(\mathbf{K})}.$$

*Proof.* Apply the singular value decomposition on $\mathbf{K}$ as $\mathbf{K} = \mathbf{U}\mathbf{D}\mathbf{Q}^T$, where $\mathbf{U} \in \mathbb{R}^{p \times p}$ and $\mathbf{Q} \in \mathbb{R}^{n \times n}$ are the orthonormal matrices, and $\mathbf{D} \in \mathbb{R}^{p \times n}$ is a rectangular diagonal matrix whose diagonal elements are $d_1, d_2, \cdots, d_p$. Then, we obtain

$$\mathbf{K}\mathbf{K}^T = \mathbf{U}\mathbf{D}\mathbf{Q}^T\mathbf{Q}\mathbf{D}^T\mathbf{U}^T = \mathbf{U}\mathbf{D}\mathbf{D}^T\mathbf{U}^T = \mathbf{U}\operatorname{diag}(d_1^2, d_2^2, \cdots, d_p^2)\mathbf{U}^T,$$

$$(\mathbf{K}\mathbf{K}^T)^{-1} = \mathbf{U}\operatorname{diag}(d_1^{-2}, d_2^{-2}, \cdots, d_p^{-2})\mathbf{U}^T,$$

and

$$\begin{aligned}
\left\|(\mathbf{K}\mathbf{K}^T)^{-1}\mathbf{K}\boldsymbol{a}\right\|^2 &= \boldsymbol{a}^T\mathbf{K}^T(\mathbf{K}\mathbf{K}^T)^{-1}(\mathbf{K}\mathbf{K}^T)^{-1}\mathbf{K}\boldsymbol{a} \\
&= \boldsymbol{a}^T\mathbf{Q}\mathbf{D}^T\operatorname{diag}(d_1^{-4}, d_2^{-4}, \cdots, d_p^{-4})\mathbf{D}\mathbf{Q}^T\boldsymbol{a} \\
&= \boldsymbol{a}^T\mathbf{Q}\operatorname{diag}(d_1^{-2}, d_2^{-2}, \cdots, d_p^{-2}, 0, \cdots, 0)\mathbf{Q}^T\boldsymbol{a}.
\end{aligned}$$

Notice that $\left\|\mathbf{Q}^T\boldsymbol{a}\right\| = \|\boldsymbol{a}\|$. Therefore, we have

$$\begin{aligned}
\left\|(\mathbf{K}\mathbf{K}^T)^{-1}\mathbf{K}\boldsymbol{a}\right\|^2 &\leq \max\left\{d_1^{-2}, d_2^{-2}, \cdots, d_p^{-2}, 0, \cdots, 0\right\} \cdot \left\|\mathbf{Q}^T\boldsymbol{a}\right\|^2 \\
&= \min\{d_1, d_2, \cdots, d_p\}^{-2} \cdot \|\boldsymbol{a}\|^2 \\
&= \frac{\|\boldsymbol{a}\|^2}{\lambda_{\min}^2(\mathbf{K})}.
\end{aligned}$$

$\square$

## B  PROOF OF THEOREM 1

Define

$$\mathbf{U}_{(1)} := \begin{bmatrix} \mathbf{X}_{(1)} \\ \mathbf{Z}_{(1)} \end{bmatrix} \in \mathbb{R}^{(p+p_{(1)}) \times n_{(1)}}, \tag{29}$$

$$\mathbf{D} := \begin{bmatrix} \mathbf{I}_p & \mathbf{0} \\ \mathbf{0} & \mathbf{0} \end{bmatrix} \in \mathbb{R}^{(p+p_{(1)}) \times (p_1+p_2)}, \tag{30}$$

$$\mathbf{G} := \mathbf{I}_{p_1+p_2} - \mathbf{D} = \begin{bmatrix} \mathbf{0} & \mathbf{0} \\ \mathbf{0} & \mathbf{I}_p \end{bmatrix} \in \mathbb{R}^{(p_1+p_2) \times (p_1+p_2)}. \tag{31}$$

When $p_1 + p_2 > n_{(1)}$ (overparameterized), we have (with probability 1)

$$\tilde{\boldsymbol{w}}_{(1),e} = \mathbf{D}\mathbf{U}_{(1)}\left(\mathbf{U}_{(1)}^T\mathbf{U}_{(1)}\right)^{-1}\left(\mathbf{X}_{(1)}^T\boldsymbol{w}_{(1)} + \mathbf{Z}_{(1)}^T\boldsymbol{q}_{(1)} + \boldsymbol{\epsilon}_{(1)}\right). \tag{32}$$

THE OVERPARAMETERIZED SITUATION

We define

$$\mathbf{P} := \mathbf{U}_{(1)}(\mathbf{U}_{(1)}^T\mathbf{U}_{(1)})^{-1}\mathbf{U}_{(1)}^T. \tag{33}$$

Notice that

$$\mathbf{P}\mathbf{P} = \mathbf{P}, \text{ and } \mathbf{P}^T = \mathbf{P}, \tag{34}$$

i.e., $\mathbf{P}$ is a projection to the column space of $\mathbf{U}_{(1)}$.

Define the extended vector of $\boldsymbol{w}_{(1)}$, $\boldsymbol{w}_{(2)}$, and $\tilde{\boldsymbol{w}}_{(1)}$ as

$$\boldsymbol{w}_{(1),e} := \begin{bmatrix} \boldsymbol{w}_{(1)} \\ \mathbf{0} \end{bmatrix} \in \mathbb{R}^{p+p_{(1)}}, \quad \boldsymbol{w}_{(2),e} := \begin{bmatrix} \boldsymbol{w}_{(2)} \\ \mathbf{0} \end{bmatrix} \in \mathbb{R}^{p+p_{(1)}}, \quad \tilde{\boldsymbol{w}}_{(1),e} := \begin{bmatrix} \tilde{\boldsymbol{w}}_{(1)} \\ \mathbf{0} \end{bmatrix} \in \mathbb{R}^{p+p_{(1)}}, \tag{35}$$

respectively. Similarly, define the extended vector of $\boldsymbol{q}_{(1)}$ as

$$\boldsymbol{q}_{(1),e} := \begin{bmatrix} \mathbf{0} \\ \boldsymbol{q}_{(1)} \end{bmatrix} \in \mathbb{R}^{p+p_{(1)}}. \tag{36}$$

By Eq. (32) and Eq. (35), we have

$$\left\|\boldsymbol{w}_{(2)} - \tilde{\boldsymbol{w}}_{(1)}\right\|^2 = \left\|\boldsymbol{w}_{(2),e} - \mathbf{D}\mathbf{U}_{(1)}\left(\mathbf{U}_{(1)}^T\mathbf{U}_{(1)}\right)^{-1}\left(\mathbf{X}_{(1)}^T\boldsymbol{w}_{(1)} + \mathbf{Z}_{(1)}^T\boldsymbol{q}_{(1)} + \boldsymbol{\epsilon}_{(1)}\right)\right\|^2$$

$$= \left\|\boldsymbol{w}_{(2),e} - \mathbf{D}\mathbf{U}_{(1)}\left(\mathbf{U}_{(1)}^T\mathbf{U}_{(1)}\right)^{-1}\left(\mathbf{U}_{(1)}^T\left[\begin{smallmatrix}\boldsymbol{w}_{(1)}\\\boldsymbol{q}_{(1)}\end{smallmatrix}\right] + \boldsymbol{\epsilon}_{(1)}\right)\right\|^2 \quad \text{(by Eq. (29))}.$$

By Assumption 1 (noise has zero mean), we obtain

$$\mathbb{E}_{\boldsymbol{\epsilon}_{(1)}}\left\langle \boldsymbol{w}_{(2),e} - \mathbf{D}\mathbf{U}_{(1)}(\mathbf{U}_{(1)}^T\mathbf{U}_{(1)})^{-1}\mathbf{U}_{(1)}^T\left[\begin{smallmatrix}\boldsymbol{w}_{(1)}\\\boldsymbol{q}_{(1)}\end{smallmatrix}\right],\ \mathbf{D}\mathbf{U}_{(1)}(\mathbf{U}_{(1)}^T\mathbf{U}_{(1)})^{-1}\boldsymbol{\epsilon}_{(1)}\right\rangle = 0,$$

and therefore conclude

$$\mathbb{E}_{\boldsymbol{\epsilon}_{(1)}}\left\|\boldsymbol{w}_{(2)} - \tilde{\boldsymbol{w}}_{(1)}\right\|^2 = \underbrace{\left\|\boldsymbol{w}_{(2),e} - \mathbf{D}\mathbf{U}_{(1)}(\mathbf{U}_{(1)}^T\mathbf{U}_{(1)})^{-1}\mathbf{U}_{(1)}^T\left[\begin{smallmatrix}\boldsymbol{w}_{(1)}\\\boldsymbol{q}_{(1)}\end{smallmatrix}\right]\right\|^2}_{\text{Term 1}} + \underbrace{\left\|\mathbf{D}\mathbf{U}_{(1)}(\mathbf{U}_{(1)}^T\mathbf{U}_{(1)})^{-1}\boldsymbol{\epsilon}_{(1)}\right\|^2}_{\text{Term 2}}.$$

$$(37)$$

The following lemma shows the expected value of Term 2 of Eq. (37).

**Lemma 15.** *We have*

$$\mathbb{E}_{\mathbf{U}_{(1)},\boldsymbol{\epsilon}}\left[\textit{Term 2 of Eq. (37)}\right] = \frac{p}{p + p_{(1)}} \cdot \frac{n_{(1)}\sigma_{(1)}^2}{p + p_{(1)} - n_{(1)} - 1}.$$

*Proof.* See Appendix B.2. □

It remains to estimate Term 1 of Eq. (37). The following lemma aims to achieve a relatively precise estimation when $r\left(\left\|\boldsymbol{w}_{(1)}\right\|^2 + \left\|\boldsymbol{q}_{(1)}\right\|^2\right)$ is relatively small with respect to $\delta$.

**Lemma 16.** *We have*

$$\left[\delta - \sqrt{r\left(\left\|\boldsymbol{w}_{(1)}\right\|^2 + \left\|\boldsymbol{q}_{(1)}\right\|^2\right)}\right]_+^2 \leq \mathbb{E}[\textit{Term 1 in Eq. (37)}] \leq \bar{b}_1^2.$$

*Proof.* See Appendix B.3. □

However, Lemma 16 may be loose in some other cases, which requires some finer estimation on Term 1 of Eq. (37) for those cases. Define

$$A := \boldsymbol{w}_{(2),e} - \boldsymbol{w}_{(1),e}, \tag{38}$$

$$B := \boldsymbol{w}_{(1),e} - \mathbf{D}\mathbf{U}_{(1)}(\mathbf{U}_{(1)}^T\mathbf{U}_{(1)})^{-1}\mathbf{X}_{(1)}^T\boldsymbol{w}_{(1)}, \tag{39}$$

$$C := \mathbf{D}\mathbf{U}_{(1)}(\mathbf{U}_{(1)}^T\mathbf{U}_{(1)})^{-1}\mathbf{Z}_{(1)}^T\boldsymbol{q}_{(1)}. \tag{40}$$

We then obtain

$$\text{Term 1 in Eq. (37)} = \|A + B + C\|^2 \tag{41}$$

The following lemmas concern the estimation of $A$, $B$, and $C$.

**Lemma 17.** *We have*

$$\left[\left\|\boldsymbol{w}_{(2)}\right\| - \sqrt{1-r}\left\|\boldsymbol{w}_{(1)}\right\|\right]_+^2 \leq \mathbb{E}\left[\left\|A + B\right\|^2\right] \leq \left(\left\|\boldsymbol{w}_{(2)}\right\| + \sqrt{1-r}\left\|\boldsymbol{w}_{(1)}\right\|\right)^2.$$

*Proof.* See Appendix B.4 □

**Lemma 18.** *We have*

$$\left[1 - \frac{2n_{(1)}}{p + p_{(1)}}\right]_+ \cdot \left\|\boldsymbol{w}_{(1)}\right\|^2 \leq \mathbb{E}\left[\left\|B\right\|^2\right] \leq r\left\|\boldsymbol{w}_{(1)}\right\|^2.$$

*Proof.* See Appendix B.5. □

**Lemma 19.** *We have*

$$\mathbb{E}\left[\|C\|^2\right] \leq \min\{r, 1-r\} \left\|\boldsymbol{q}_{(1)}\right\|^2.$$

*Proof.* See Appendix B.6. □

We therefore obtain

$$\mathbb{E}[\text{Term 1 of Eq. (37)}]$$
$$= \mathbb{E}[\|A + B + C\|^2] \text{ (by Eq. (41))}$$
$$\leq \left(\sqrt{\mathbb{E}[\|A+B\|^2]} + \sqrt{\mathbb{E}[\|C\|^2]}\right)^2 \text{ (by Lemma 6)} \tag{42}$$
$$\leq \left(\left\|\boldsymbol{w}_{(2)}\right\| + \sqrt{1-r}\left\|\boldsymbol{w}_{(1)}\right\| + \sqrt{\min\{r,1-r\}}\left\|\boldsymbol{q}_{(1)}\right\|\right)^2 \text{ (by Lemma 17 and Lemma 19)}$$
$$= \bar{b}_2^2, \tag{43}$$

and

$$\mathbb{E}[\text{Term 1 of Eq. (37)}]$$
$$= \mathbb{E}[\|A + B + C\|^2] \text{ (by Eq. (41))}$$
$$\geq \left[\sqrt{\mathbb{E}[\|A+B\|^2]} - \sqrt{\mathbb{E}[\|C\|^2]}\right]_+^2 \text{ (by Lemma 6)} \tag{44}$$
$$\geq \left[\left\|\boldsymbol{w}_{(2)}\right\| - \sqrt{1-r}\left\|\boldsymbol{w}_{(1)}\right\| - \sqrt{\min\{r,1-r\}}\left\|\boldsymbol{q}_{(1)}\right\|\right]_+^2 \text{ (by Lemma 17 and Lemma 19)}.$$

We also have

$$\mathbb{E}[\text{Term 1 of Eq. (37)}]$$
$$\leq \left(\sqrt{\mathbb{E}[\|A+B\|^2]} + \sqrt{\mathbb{E}[\|C\|^2]}\right)^2 \text{ (by Eq. (42))}$$
$$\leq \left(\sqrt{\mathbb{E}[\|A\|^2]} + \sqrt{\mathbb{E}[\|B\|^2]} + \sqrt{\mathbb{E}[\|C\|^2]}\right)^2 \text{ (by Lemma 6)}$$
$$\leq \left(\delta + \sqrt{r}\left\|\boldsymbol{w}_{(1)}\right\| + \sqrt{\min\{r,1-r\}}\left\|\boldsymbol{q}_{(1)}\right\|\right)^2 \text{ (by Eq. (7), Lemma 18, and Lemma 19)}$$
$$= \bar{b}_3^2, \tag{45}$$

and

$$\mathbb{E}[\text{Term 1 of Eq. (37)}]$$
$$\geq \left[\sqrt{\mathbb{E}[\|A+B\|^2]} - \sqrt{\mathbb{E}[\|C\|^2]}\right]_+^2 \text{ (by Eq. (44))}$$
$$\geq \left[\sqrt{\mathbb{E}[\|B\|^2]} - \sqrt{\mathbb{E}[\|A\|^2]} - \sqrt{\mathbb{E}[\|C\|^2]}\right]_+^2 \text{ (by Lemma 6)}$$
$$\geq \left[\sqrt{r}\left\|\boldsymbol{w}_{(1)}\right\| - \delta - \sqrt{\min\{r,1-r\}}\left\|\boldsymbol{q}_{(1)}\right\|\right]_+^2 \text{ (by Eq. (7), Lemma 18, and Lemma 19)}.$$

By Lemma 16, Eq. (43), and Eq. (45), we therefore conclude

$$\mathbb{E}[\text{Term 1 of Eq. (37)}] \leq \min_{i=1,2,3} \bar{b}_i^2. \tag{46}$$

By Eq. (46), Lemma 15, and Eq. (37), we thus have Eq. (8).

### THE UNDERPARAMETERIZED SITUATION

When $p + p_{(1)} < n_{(1)}$, the solution that minimizes the training error is given by

$$
\begin{aligned}
\tilde{w}_{(1),e} =& \mathbf{D} \left( \mathbf{U}_{(1)} \mathbf{U}_{(1)}^T \right)^{-1} \mathbf{U}_{(1)} \boldsymbol{y}_{(1)} \\
=& \mathbf{D} \left( \mathbf{U}_{(1)} \mathbf{U}_{(1)}^T \right)^{-1} \mathbf{U}_{(1)} \left( \mathbf{X}_{(1)}^T \boldsymbol{w}_{(1)} + \mathbf{Z}_{(1)}^T \boldsymbol{q}_{(1)} + \boldsymbol{\epsilon}_{(1)} \right) \\
=& \mathbf{D} \left( \mathbf{U}_{(1)} \mathbf{U}_{(1)}^T \right)^{-1} \mathbf{U}_{(1)} \left( \mathbf{U}_{(1)}^T \left[ \begin{smallmatrix} \boldsymbol{w}_{(1)} \\ \boldsymbol{q}_{(1)} \end{smallmatrix} \right] + \boldsymbol{\epsilon}_{(1)} \right) \\
=& \boldsymbol{w}_{(1),e} + \mathbf{D} \left( \mathbf{U}_{(1)} \mathbf{U}_{(1)}^T \right)^{-1} \mathbf{U}_{(1)} \boldsymbol{\epsilon}_{(1)} \text{ (by Eq. (35) and Eq. (30)).}
\end{aligned}
$$

Thus, we have

$$
\left\| \boldsymbol{w}_{(2)} - \tilde{w}_{(1)} \right\|^2 = \left\| (\boldsymbol{w}_{(2),e} - \boldsymbol{w}_{(1),e}) - \mathbf{D} \left( \mathbf{U}_{(1)} \mathbf{U}_{(1)}^T \right)^{-1} \mathbf{U}_{(1)} \boldsymbol{\epsilon}_{(1)} \right\|^2. \tag{47}
$$

By Assumption 1, we know that $\boldsymbol{\epsilon}_{(1)}$ is independent of $\mathbf{U}_{(1)}$ and has zero mean. We therefore obtain

$$
\mathbb{E}_{\boldsymbol{\epsilon}_{(1)}} \left\langle \boldsymbol{w}_{(2),e} - \tilde{w}_{(1),e}, \mathbf{D} \left( \mathbf{U}_{(1)} \mathbf{U}_{(1)}^T \right)^{-1} \mathbf{U}_{(1)} \boldsymbol{\epsilon}_{(1)} \right\rangle = 0. \tag{48}
$$

Recalling the definition of $\delta$ in Eq. (7), by Eq. (47) and Eq. (48), we thus have

$$
\mathbb{E} \left\| \boldsymbol{w}_{(2)} - \tilde{w}_{(1)} \right\|^2 = \delta^2 + \mathbb{E} \left\| \mathbf{D} \left( \mathbf{U}_{(1)} \mathbf{U}_{(1)}^T \right)^{-1} \mathbf{U}_{(1)} \boldsymbol{\epsilon}_{(1)} \right\|^2. \tag{49}
$$

It remains to estimate $\mathbb{E} \left\| \mathbf{D} \left( \mathbf{U}_{(1)} \mathbf{U}_{(1)}^T \right)^{-1} \mathbf{U}_{(1)} \boldsymbol{\epsilon}_{(1)} \right\|^2$. To that end, we derive

$$
\begin{aligned}
& \mathbb{E} \left\| \mathbf{D} \left( \mathbf{U}_{(1)} \mathbf{U}_{(1)}^T \right)^{-1} \mathbf{U}_{(1)} \boldsymbol{\epsilon}_{(1)} \right\|^2 \\
=& \mathbb{E} \left[ \left( \mathbf{D} \left( \mathbf{U}_{(1)} \mathbf{U}_{(1)}^T \right)^{-1} \mathbf{U}_{(1)} \boldsymbol{\epsilon}_{(1)} \right)^T \mathbf{D} \left( \mathbf{U}_{(1)} \mathbf{U}_{(1)}^T \right)^{-1} \mathbf{U}_{(1)} \boldsymbol{\epsilon}_{(1)} \right] \\
=& \mathbb{E} \operatorname{Tr} \left( \left( \mathbf{D} \left( \mathbf{U}_{(1)} \mathbf{U}_{(1)}^T \right)^{-1} \mathbf{U}_{(1)} \boldsymbol{\epsilon}_{(1)} \right)^T \mathbf{D} \left( \mathbf{U}_{(1)} \mathbf{U}_{(1)}^T \right)^{-1} \mathbf{U}_{(1)} \boldsymbol{\epsilon}_{(1)} \right) \\
=& \mathbb{E} \operatorname{Tr} \left( \mathbf{D} \left( \mathbf{U}_{(1)} \mathbf{U}_{(1)}^T \right)^{-1} \mathbf{U}_{(1)} \boldsymbol{\epsilon}_{(1)} \boldsymbol{\epsilon}_{(1)}^T \mathbf{U}_{(1)}^T (\mathbf{U}_{(1)} \mathbf{U}_{(1)}^T)^{-1} \mathbf{D} \right) \\
& \text{(by trace trick that } \operatorname{Tr}(\mathbf{AB}) = \operatorname{Tr}(\mathbf{BA})\text{)} \\
=& \mathbb{E}_{\mathbf{U}_{(1)}} \operatorname{Tr} \left( \mathbf{D} \left( \mathbf{U}_{(1)} \mathbf{U}_{(1)}^T \right)^{-1} \mathbf{U}_{(1)} \mathbb{E}_{\boldsymbol{\epsilon}_{(1)}} [\boldsymbol{\epsilon}_{(1)} \boldsymbol{\epsilon}_{(1)}^T] \mathbf{U}_{(1)}^T (\mathbf{U}_{(1)} \mathbf{U}_{(1)}^T)^{-1} \mathbf{D} \right) \\
=& \sigma_{(1)}^2 \mathbb{E} \operatorname{Tr} \left( \mathbf{D} (\mathbf{U}_{(1)} \mathbf{U}_{(1)}^T)^{-1} \mathbf{D} \right) \text{ (} \mathbb{E}[\boldsymbol{\epsilon}_{(1)} \boldsymbol{\epsilon}_{(1)}^T] = \sigma_{(1)}^2 \mathbf{I}_{n_{(1)}} \text{ by Assumption 1)} \\
=& \operatorname{Tr} \left( \mathbf{D} \, \mathbb{E}[(\mathbf{U}_{(1)} \mathbf{U}_{(1)}^T)^{-1}] \mathbf{D} \right) \\
=& \frac{p}{p + p_{(1)}} \cdot \frac{(p + p_{(1)}) \sigma_{(1)}^2}{n_{(1)} - (p + p_{(1)}) - 1} \text{ (by Lemma 8)} \\
=& \frac{p \sigma_{(1)}^2}{n_{(1)} - (p + p_{(1)}) - 1}. \tag{50}
\end{aligned}
$$

Substituting Eq. (50) into Eq. (49), we therefore obtain

$$
\mathbb{E} \left\| \boldsymbol{w}_{(2)} - \tilde{w}_{(1)} \right\|^2 = \delta^2 + \frac{p \sigma_{(1)}^2}{n_{(1)} - (p + p_{(1)}) - 1},
$$

i.e., we have Eq. (9).

Combining our proof for the overparameterized situation and the underparameterized situation, Theorem 1 thus follows.

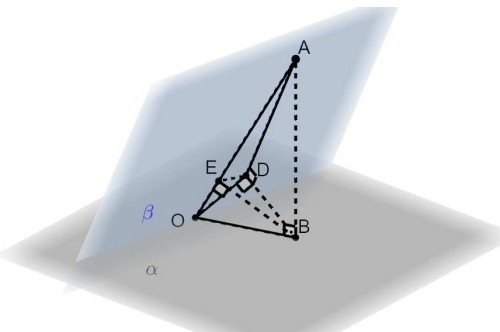

Figure 2: Geometric illustration of some vectors and projections used in the proof.

| Notation in Fig. 2 | Meaning |
|---|---|
| plane $\alpha$ | column space of $\mathbf{U}_{(1)}$ |
| plane $\beta$ | subspace spanned by $\boldsymbol{e}_1, \cdots, \boldsymbol{e}_p$ |
| vector $\overrightarrow{OA}$ | $\boldsymbol{w}_{(1),e}$ |
| vector $\overrightarrow{OB}$ | $\mathbf{P}\boldsymbol{w}_{(1),e}$ |
| vector $\overrightarrow{OD}$ | $\mathbf{D}\mathbf{P}\boldsymbol{w}_{(1),e}$ |
| vector $\overrightarrow{DA}$ | $\boldsymbol{w}_{(1),e} - \mathbf{D}\mathbf{P}\boldsymbol{w}_{(1),e}$ |
| vector $\overrightarrow{OE}$ | $\langle \boldsymbol{w}_{(1),e}, \, \mathbf{D}\mathbf{P}\boldsymbol{w}_{(1),e}\rangle \cdot \boldsymbol{w}_{(1),e} / \left\|\boldsymbol{w}_{(1),e}\right\|^2$ |
| vector $\overrightarrow{EA}$ | $\boldsymbol{w}_{(1),e} - \langle \boldsymbol{w}_{(1),e}, \, \mathbf{D}\mathbf{P}\boldsymbol{w}_{(1),e}\rangle \cdot \boldsymbol{w}_{(1),e} / \left\|\boldsymbol{w}_{(1),e}\right\|^2$ |

Table 1: Interpretation of notations in Fig. 2 for Lemma 18.

### B.1 TIGHTNESS OF EQ. (8)

All our estimations except $\mathcal{L}_{\text{co}}^{\text{noiseless}}$ are precise. It remains to check the tightness of Eq. (8), i.e., $\min_{i=1,2,3} \bar{b}_i^2$. Here $\bar{b}_1, \bar{b}_2$, and $\bar{b}_3$ are for different situations.

First, when $r$ and $\delta$ are small, $\bar{b}_1$ is the smallest. In this case, $\bar{b}_1 \leq \bar{b}_3$ because $\sqrt{r\left(\left\|\boldsymbol{w}_{(1)}\right\|^2 + \left\|\boldsymbol{q}_{(1)}\right\|^2\right)} \leq \sqrt{r}\left\|\boldsymbol{w}_{(1)}\right\| + \sqrt{r}\left\|\boldsymbol{q}_{(1)}\right\|$; and $\bar{b}_1 \leq \bar{b}_2$ because $r = \left\|\boldsymbol{w}_{(2)} - \boldsymbol{w}_{(1)}\right\| \leq \left\|\boldsymbol{w}_{(2)}\right\| + \left\|\boldsymbol{w}_{(1)}\right\| \approx \left\|\boldsymbol{w}_{(2)}\right\| + \left\|\boldsymbol{w}_{(1)}\right\|$ when $r$ is small. In Fig. 3(a)(c) (where $\delta = 0$), the curves of $\bar{b}_1^2$ (blue dashed curves) are the closest ones to the empirical value of $\mathcal{L}_{\text{co}}^{\text{noiseless}}$ (black curves with markers "+") in the near overparameterized regime.

Second, if only $\delta$ is small but $r$ is large, then $\bar{b}_3$ is the smallest. In this case, $\bar{b}_3 \leq \bar{b}_1$ because the coefficient of $\left\|\boldsymbol{q}_{(1)}\right\|$ in $\bar{b}_3$ approaches 0 when $r \to 1$ while that in $\bar{b}_1$ approaches 1. For this reason, the difference between $\bar{b}_3$ and $\bar{b}_1$ can be observed more easily when $\left\|\boldsymbol{q}_{(1)}\right\|$ is large. In Fig. 3(a)(c), we can see that the curves of $\bar{b}_3^2$ are the closest ones to the actual value of $\mathcal{L}_{\text{co}}^{\text{noiseless}}$ when $p$ is large (i.e., $r$ is large). As expected, the difference between $\bar{b}_3$ and $\bar{b}_1$ in Fig. 3(b) is larger and easier to observe compared to that in Fig. 3(a).

Third, when $\delta$ is large, $\bar{b}_2$ can sometimes be the smallest since $\boldsymbol{w}_{(2)}$ and $\boldsymbol{w}_{(1)}$ are treated separately in the expression of $\bar{b}_2$ to get a more precise estimation. In Fig. 3(b)(d) (where $\delta$ is very large), the curves of $\bar{b}_2^2$ (red dashed ones) are the closest ones to the curve of $\mathcal{L}_{\text{co}}^{\text{noiseless}}$ (black curves with the markers "+").

| Notation in Fig. 2 | Meaning |
|---|---|
| plane $\alpha$ | column space of $\mathbf{U}_{(1)}$ |
| plane $\beta$ | subspace spanned by $\boldsymbol{e}_{p+1}, \cdots, \boldsymbol{e}_{p+p_{(1)}}$ |
| vector $\overrightarrow{OA}$ | $\boldsymbol{q}_{(1),e}$ |
| vector $\overrightarrow{OB}$ | $\mathbf{P}\boldsymbol{q}_{(1),e}$ |
| vector $\overrightarrow{OD}$ | $\mathbf{G}\mathbf{P}\boldsymbol{q}_{(1),e}$ |
| vector $\overrightarrow{DB}$ | $\mathbf{D}\mathbf{P}\boldsymbol{q}_{(1),e}$ |

Table 2: Interpretation of notations in Fig. 2 for Lemma 19.

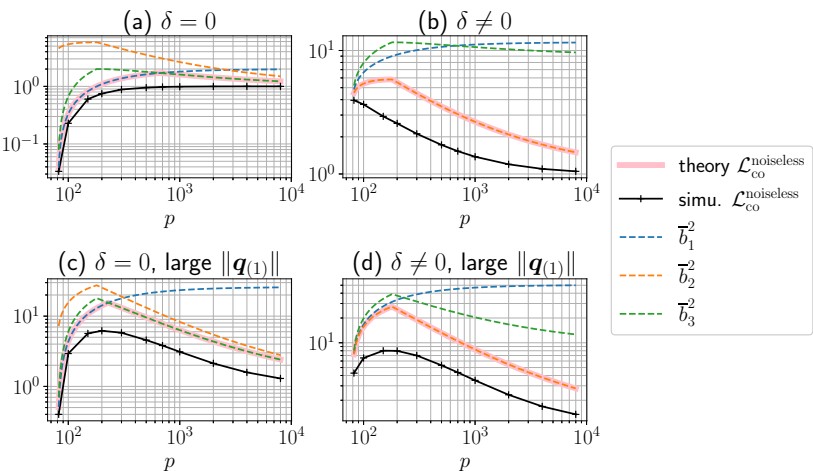

Figure 3: The curves of $\mathcal{L}_{\text{co}}^{\text{noiseless}}$ with respect to $p$ in the overparameterized region, where $n_{(1)} = 100$, $\left\|\boldsymbol{w}_{(1)}\right\| = 1$, and $p_{(1)} = 20$. Each data point is the average of 20 runs with different random seeds (randomness is on input features). Some other settings for each subfigure: (a) $\boldsymbol{w}_{(1)} = \boldsymbol{w}_{(2)}$ and $\left\|\boldsymbol{q}_{(1)}\right\| = 1$; (b) $\boldsymbol{w}_{(1)} = -\boldsymbol{w}_{(2)}$, and $\left\|\boldsymbol{q}_{(1)}\right\| = 1$; (c) $\boldsymbol{w}_{(1)} = \boldsymbol{w}_{(2)}$ and $\left\|\boldsymbol{q}_{(1)}\right\| = 5$; (d) $\boldsymbol{w}_{(1)} = -\boldsymbol{w}_{(2)}$, and $\left\|\boldsymbol{q}_{(1)}\right\| = 5$.

### B.2 PROOF OF LEMMA 15

*Proof.* We first prove that $\mathbf{U}_{(1)}(\mathbf{U}_{(1)}^T\mathbf{U}_{(1)})^{-1}\boldsymbol{\epsilon}_{(1)}$ has rotation symmetry. Consider any rotation $\mathbf{S} \in \mathsf{SO}(p + p_{(1)})$ where $\mathsf{SO}(p + p_{(1)}) \in \mathbb{R}^{(p+p_{(1)}) \times (p+p_{(1)})}$ denotes all rotations in $(p + p_{(1)})$-dimensional space. We have

$$
\begin{aligned}
\mathbf{S}\mathbf{U}_{(1)}(\mathbf{U}_{(1)}^T\mathbf{U}_{(1)})^{-1}\boldsymbol{\epsilon}_{(1)} &= \mathbf{S}\mathbf{U}_{(1)}(\mathbf{U}_{(1)}^T\mathbf{S}^{-1}\mathbf{S}\mathbf{U}_{(1)})^{-1}\boldsymbol{\epsilon}_{(1)} \\
&= \mathbf{S}\mathbf{U}_{(1)}(\mathbf{U}_{(1)}^T\mathbf{S}^T\mathbf{S}\mathbf{U}_{(1)})^{-1}\boldsymbol{\epsilon}_{(1)} \ (\mathbf{S}^{-1} = \mathbf{S}^T \text{ since } \mathbf{S} \text{ is a rotation}) \\
&= (\mathbf{S}\mathbf{U}_{(1)})((\mathbf{S}\mathbf{U}_{(1)})^T\mathbf{S}\mathbf{U}_{(1)})^{-1}\boldsymbol{\epsilon}_{(1)}.
\end{aligned}
$$

By Assumption 1, we know that $\mathbf{S}\mathbf{U}_{(1)}$ has the same distribution as $\mathbf{U}_{(1)}$. Thus, the rotated vector $\mathbf{S}\mathbf{U}_{(1)}(\mathbf{U}_{(1)}^T\mathbf{U}_{(1)})^{-1}\boldsymbol{\epsilon}_{(1)}$ has the same distribution as the original vector $\mathbf{U}_{(1)}(\mathbf{U}_{(1)}^T\mathbf{U}_{(1)})^{-1}\boldsymbol{\epsilon}_{(1)}$. This implies that

$$
\mathbb{E}([\mathbf{U}_{(1)}(\mathbf{U}_{(1)}^T\mathbf{U}_{(1)})^{-1}\boldsymbol{\epsilon}_{(1)}]_1)^2 = \mathbb{E}([\mathbf{U}_{(1)}(\mathbf{U}_{(1)}^T\mathbf{U}_{(1)})^{-1}\boldsymbol{\epsilon}_{(1)}]_2)^2 = \cdots = \mathbb{E}([\mathbf{U}_{(1)}(\mathbf{U}_{(1)}^T\mathbf{U}_{(1)})^{-1}\boldsymbol{\epsilon}_{(1)}]_{p+p_{(1)}})^2,
$$

where $[\cdot]_i$ denotes the $i$-th element of the vector. We therefore obtain

$$
\begin{aligned}
\mathbb{E}\left\|\mathbf{D}\mathbf{U}_{(1)}(\mathbf{U}_{(1)}^T\mathbf{U}_{(1)})^{-1}\boldsymbol{\epsilon}_{(1)}\right\|^2 &= \sum_{i=1}^{p}\mathbb{E}([\mathbf{U}_{(1)}(\mathbf{U}_{(1)}^T\mathbf{U}_{(1)})^{-1}\boldsymbol{\epsilon}_{(1)}]_i)^2 \\
&= \frac{p}{p+p_{(1)}}\,\mathbb{E}\left\|\mathbf{U}_{(1)}(\mathbf{U}_{(1)}^T\mathbf{U}_{(1)})^{-1}\boldsymbol{\epsilon}_{(1)}\right\|^2 \\
&= \frac{p}{p+p_{(1)}}\cdot\frac{n_{(1)}\sigma_{(1)}^2}{p+p_{(1)}-n_{(1)}-1}\ \text{(by Lemma 8)}.
\end{aligned}
$$

The result of this lemma thus follows. $\qquad\square$

### B.3 PROOF OF LEMMA 16

*Proof.* By Eq. (30) and Eq. (35), we have

$$
\boldsymbol{w}_{(2),e} = \mathbf{D}\begin{bmatrix}\boldsymbol{w}_{(2)}\\\boldsymbol{q}_{(1)}\end{bmatrix} = \mathbf{D}\begin{bmatrix}\boldsymbol{w}_{(2)}-\boldsymbol{w}_{(1)}\\\mathbf{0}\end{bmatrix} + \mathbf{D}\begin{bmatrix}\boldsymbol{w}_{(1)}\\\boldsymbol{q}_{(1)}\end{bmatrix} = \begin{bmatrix}\boldsymbol{w}_{(2)}\\\boldsymbol{q}_{(1)}\end{bmatrix} = \mathbf{D}\begin{bmatrix}\boldsymbol{w}_{(2)}-\boldsymbol{w}_{(1)}\\\mathbf{0}\end{bmatrix} + \mathbf{D}\begin{bmatrix}\boldsymbol{w}_{(1)}\\\boldsymbol{q}_{(1)}\end{bmatrix}. \quad (51)
$$

Thus, we obtain

$$
\mathbb{E}[\text{Term 1 of Eq. (37)}]
$$

$$
= \mathbb{E}\left\|\mathbf{D}\left(\begin{bmatrix}\boldsymbol{w}_{(2)}-\boldsymbol{w}_{(1)}\\\mathbf{0}\end{bmatrix} + \begin{bmatrix}\boldsymbol{w}_{(1)}\\\boldsymbol{q}_{(1)}\end{bmatrix} - \mathbf{U}_{(1)}(\mathbf{U}_{(1)}^T\mathbf{U}_{(1)})^{-1}\mathbf{U}_{(1)}^T\begin{bmatrix}\boldsymbol{w}_{(1)}\\\boldsymbol{q}_{(1)}\end{bmatrix}\right)\right\|^2\ \text{(by Eq. (51))}
$$

$$
\leq \mathbb{E}\left\|\begin{bmatrix}\boldsymbol{w}_{(2)}-\boldsymbol{w}_{(1)}\\\mathbf{0}\end{bmatrix} + \begin{bmatrix}\boldsymbol{w}_{(1)}\\\boldsymbol{q}_{(1)}\end{bmatrix} - \mathbf{U}_{(1)}(\mathbf{U}_{(1)}^T\mathbf{U}_{(1)})^{-1}\mathbf{U}_{(1)}^T\begin{bmatrix}\boldsymbol{w}_{(1)}\\\boldsymbol{q}_{(1)}\end{bmatrix}\right\|^2\ \text{(since } \|\mathbf{D}\boldsymbol{a}\|\leq\|\boldsymbol{a}\|\text{ for all }\boldsymbol{a}\in\mathbb{R}^{p+p_{(1)}})
$$

$$
\leq \left(\delta + \sqrt{\mathbb{E}\left\|\left(\begin{bmatrix}\boldsymbol{w}_{(1)}\\\boldsymbol{q}_{(1)}\end{bmatrix} - \mathbf{U}_{(1)}(\mathbf{U}_{(1)}^T\mathbf{U}_{(1)})^{-1}\mathbf{U}_{(1)}^T\begin{bmatrix}\boldsymbol{w}_{(1)}\\\boldsymbol{q}_{(1)}\end{bmatrix}\right)\right\|^2}\right)^2\ \text{(by Lemma 6 and Eq. (7))}
$$

$$
= \left(\delta + \sqrt{r\left(\|\boldsymbol{w}_{(1)}\|^2 + \|\boldsymbol{q}_{(1)}\|^2\right)}\right)^2\ \text{(by Lemma 7)}
$$

$$
= \bar{b}_1^2. \quad (52)
$$

We also have

$$
\mathbb{E}[\text{Term 1 of Eq. (37)}]
$$

$$
= \mathbb{E}\left\|\begin{bmatrix}\boldsymbol{w}_{(2)}-\boldsymbol{w}_{(1)}\\\mathbf{0}\end{bmatrix} + \mathbf{D}\left(\begin{bmatrix}\boldsymbol{w}_{(1)}\\\boldsymbol{q}_{(1)}\end{bmatrix} - \mathbf{U}_{(1)}(\mathbf{U}_{(1)}^T\mathbf{U}_{(1)})^{-1}\mathbf{U}_{(1)}^T\begin{bmatrix}\boldsymbol{w}_{(1)}\\\boldsymbol{q}_{(1)}\end{bmatrix}\right)\right\|^2\ \text{(by Eq. (51))}
$$

$$
\geq \left(\delta - \sqrt{\mathbb{E}\left\|\mathbf{D}\left(\begin{bmatrix}\boldsymbol{w}_{(1)}\\\boldsymbol{q}_{(1)}\end{bmatrix} - \mathbf{U}_{(1)}(\mathbf{U}_{(1)}^T\mathbf{U}_{(1)})^{-1}\mathbf{U}_{(1)}^T\begin{bmatrix}\boldsymbol{w}_{(1)}\\\boldsymbol{q}_{(1)}\end{bmatrix}\right)\right\|^2}\right)^2\ \text{(by Lemma 6 and Eq. (7))}
$$

$$
\geq \left(\left[\delta - \sqrt{\mathbb{E}\left\|\mathbf{D}\left(\begin{bmatrix}\boldsymbol{w}_{(1)}\\\boldsymbol{q}_{(1)}\end{bmatrix} - \mathbf{U}_{(1)}(\mathbf{U}_{(1)}^T\mathbf{U}_{(1)})^{-1}\mathbf{U}_{(1)}^T\begin{bmatrix}\boldsymbol{w}_{(1)}\\\boldsymbol{q}_{(1)}\end{bmatrix}\right)\right\|^2}\right]^+\right)^2
$$

$$
\geq \left[\delta - \sqrt{\mathbb{E}\left\|\begin{bmatrix}\boldsymbol{w}_{(1)}\\\boldsymbol{q}_{(1)}\end{bmatrix} - \mathbf{U}_{(1)}(\mathbf{U}_{(1)}^T\mathbf{U}_{(1)})^{-1}\mathbf{U}_{(1)}^T\begin{bmatrix}\boldsymbol{w}_{(1)}\\\boldsymbol{q}_{(1)}\end{bmatrix}\right\|^2}\right]_+^2\ \text{(since } \|\mathbf{D}\boldsymbol{a}\|\leq\|\boldsymbol{a}\|\text{ for all }\boldsymbol{a}\in\mathbb{R}^{p+p_{(1)}})
$$

$$
= \left[\delta - \sqrt{r\left(\|\boldsymbol{w}_{(1)}\|^2 + \|\boldsymbol{q}_{(1)}\|^2\right)}\right]_+^2\ \text{(by Lemma 7)}. \quad (53)
$$

By Eq. (52) and Eq. (53), the result of this lemma thus follows. $\qquad\square$

### B.4 PROOF OF LEMMA 17

*Proof.* We have

$$
\begin{aligned}
\|A+B\|^2 &= \left\|\boldsymbol{w}_{(2),e} - \mathbf{D}\mathbf{U}_{(1)}(\mathbf{U}_{(1)}^T\mathbf{U}_{(1)})^{-1}\mathbf{X}_{(1)}^T\boldsymbol{w}_{(1)}\right\|^2\ \text{(by Eq. (38) and Eq. (39))} \\
&= \left\|\boldsymbol{w}_{(2),e} - \mathbf{D}\mathbf{U}_{(1)}(\mathbf{U}_{(1)}^T\mathbf{U}_{(1)})^{-1}\mathbf{U}_{(1)}^T\boldsymbol{w}_{(1),e}\right\|^2\ \text{(by Eq. (35) and Eq. (29))}. \quad (54)
\end{aligned}
$$

Thus, we obtain

$$\mathbb{E}\|A+B\|^2 \leq \left(\|\boldsymbol{w}_{(2)}\| + \sqrt{\mathbb{E}\left\|\mathbf{D}\mathbf{U}_{(1)}(\mathbf{U}_{(1)}^T\mathbf{U}_{(1)})^{-1}\mathbf{U}_{(1)}^T\boldsymbol{w}_{(1),e}\right\|^2}\right)^2 \quad \text{(by Eq. (54) and Lemma 6)}$$

$$\leq \left(\|\boldsymbol{w}_{(2)}\| + \sqrt{\mathbb{E}\left\|\mathbf{U}_{(1)}(\mathbf{U}_{(1)}^T\mathbf{U}_{(1)})^{-1}\mathbf{U}_{(1)}^T\boldsymbol{w}_{(1),e}\right\|^2}\right)^2 \quad \text{(since } \|\mathbf{D}\boldsymbol{a}\| \leq \|\boldsymbol{a}\| \text{ for all } \boldsymbol{a}\in\mathbb{R}^{p+p_{(1)}})$$

$$= \left(\|\boldsymbol{w}_{(2)}\| + \sqrt{1-r}\,\|\boldsymbol{w}_{(1)}\|\right)^2 \quad \text{(by Lemma 7)}.$$

We also have

$$\mathbb{E}\|A+B\|^2 \geq \left(\|\boldsymbol{w}_{(2)}\| - \sqrt{\mathbb{E}\left\|\mathbf{D}\mathbf{U}_{(1)}(\mathbf{U}_{(1)}^T\mathbf{U}_{(1)})^{-1}\mathbf{U}_{(1)}^T\boldsymbol{w}_{(1),e}\right\|^2}\right)^2 \quad \text{(by Eq. (54) and Lemma 6)}$$

$$\geq \left[\|\boldsymbol{w}_{(2)}\| - \sqrt{\mathbb{E}\left\|\mathbf{U}_{(1)}(\mathbf{U}_{(1)}^T\mathbf{U}_{(1)})^{-1}\mathbf{U}_{(1)}^T\boldsymbol{w}_{(1),e}\right\|^2}\right]_+^2 \quad \text{(since } \|\mathbf{D}\boldsymbol{a}\| \leq \|\boldsymbol{a}\| \text{ for all } \boldsymbol{a}\in\mathbb{R}^{p+p_{(1)}})$$

$$= \left[\|\boldsymbol{w}_{(2)}\| - \sqrt{1-r}\,\|\boldsymbol{w}_{(1)}\|\right]_+^2 \quad \text{(by Lemma 7)}.$$

The result of this lemma thus follows. $\qquad\square$

### B.5    PROOF OF LEMMA 18

*Proof.* We first prove the upper bound in this lemma. By the definition of $\boldsymbol{w}_{(1),e}$ in Eq. (35), we obtain

$$\mathbf{X}_{(1)}^T\boldsymbol{w}_{(1)} = \mathbf{U}_{(1)}^T\boldsymbol{w}_{(1),e}, \quad \mathbf{D}\boldsymbol{w}_{(1),e} = \boldsymbol{w}_{(1),e}. \tag{55}$$

Notice that for any vector $\boldsymbol{a}\in\mathbb{R}^{p+p_{(1)}}$, we have $\|\mathbf{D}\boldsymbol{a}\| \leq \|\boldsymbol{a}\|$. Thus, we conclude

$$\left\|\boldsymbol{w}_{(1),e} - \mathbf{P}\boldsymbol{w}_{(1),e}\right\| \geq \left\|\mathbf{D}\left(\boldsymbol{w}_{(1),e} - \mathbf{P}\boldsymbol{w}_{(1),e}\right)\right\|. \tag{56}$$

By Eq. (55), we have

$$\mathbf{D}\left(\boldsymbol{w}_{(1),e} - \mathbf{P}\boldsymbol{w}_{(1),e}\right) = \boldsymbol{w}_{(1),e} - \mathbf{D}\mathbf{U}_{(1)}(\mathbf{U}_{(1)}^T\mathbf{U}_{(1)})^{-1}\mathbf{X}_{(1)}^T\boldsymbol{w}_{(1)}. \tag{57}$$

By Eq. (56) and Eq. (57), we thus obtain

$$\left\|\boldsymbol{w}_{(1),e} - \mathbf{D}\mathbf{U}_{(1)}(\mathbf{U}_{(1)}^T\mathbf{U}_{(1)})^{-1}\mathbf{X}_{(1)}^T\boldsymbol{w}_{(1)}\right\| \leq \left\|\boldsymbol{w}_{(1),e} - \mathbf{P}\boldsymbol{w}_{(1),e}\right\|. \tag{58}$$

We further have

$$\mathbb{E}\left\|\boldsymbol{w}_{(1),e} - \mathbf{D}\mathbf{U}_{(1)}(\mathbf{U}_{(1)}^T\mathbf{U}_{(1)})^{-1}\mathbf{X}_{(1)}^T\boldsymbol{w}_{(1)}\right\|^2$$

$$\leq \mathbb{E}\left\|\boldsymbol{w}_{(1),e} - \mathbf{P}\boldsymbol{w}_{(1),e}\right\|^2 \quad \text{(by Eq. (58))}$$

$$= \left(1 - \frac{n_{(1)}}{p+p_{(1)}}\right)\left\|\boldsymbol{w}_{(1),e}\right\|^2 \quad \text{(by Lemma 7)}$$

$$= \left(1 - \frac{n_{(1)}}{p+p_{(1)}}\right)\left\|\boldsymbol{w}_{(1)}\right\|^2 \quad \text{(since } \left\|\boldsymbol{w}_{(1)}\right\| = \left\|\boldsymbol{w}_{(1),e}\right\| \text{ by Eq. (35))}.$$

The upper bound in this lemma thus holds. It remains to prove the lower bound. To that end, we define

$$\boldsymbol{a} := \boldsymbol{w}_{(1),e} - \frac{\langle\mathbf{D}\mathbf{P}\boldsymbol{w}_{(1),e},\,\boldsymbol{w}_{(1),e}\rangle}{\left\|\boldsymbol{w}_{(1),e}\right\|^2}\boldsymbol{w}_{(1),e} \quad (\text{corresponds to } \overrightarrow{\mathrm{OE}} \text{ in Fig. 2}), \tag{59}$$

$$\boldsymbol{b} := \frac{\langle\mathbf{D}\mathbf{P}\boldsymbol{w}_{(1),e},\,\boldsymbol{w}_{(1),e}\rangle}{\left\|\boldsymbol{w}_{(1),e}\right\|^2}\boldsymbol{w}_{(1),e} - \mathbf{D}\mathbf{P}\boldsymbol{w}_{(1),e} \quad (\text{corresponds to } \overrightarrow{\mathrm{BE}} \text{ in Fig. 2}). \tag{60}$$

Notice that

$$\boldsymbol{a}^T\boldsymbol{b} = \langle \mathbf{DP}\boldsymbol{w}_{(1),e}, \boldsymbol{w}_{(1),e} \rangle - \langle \mathbf{DP}\boldsymbol{w}_{(1),e}, \boldsymbol{w}_{(1),e} \rangle - \frac{\langle \mathbf{DP}\boldsymbol{w}_{(1),e}, \boldsymbol{w}_{(1),e} \rangle^2}{\left\| \boldsymbol{w}_{(1),e} \right\|^2} + \frac{\langle \mathbf{DP}\boldsymbol{w}_{(1),e}, \boldsymbol{w}_{(1),e} \rangle^2}{\left\| \boldsymbol{w}_{(1),e} \right\|^2} = 0.$$
(61)

Thus, we obtain

$$\begin{aligned}
& \left\| \boldsymbol{w}_{(1),e} - \mathbf{DP}\boldsymbol{w}_{(1),e} \right\|^2 \\
&= \|\boldsymbol{a} + \boldsymbol{b}\|^2 \text{ (by Eq. (59) and Eq. (60))} \\
&= \|\boldsymbol{a}\|^2 + \|\boldsymbol{b}\|^2 \text{ (by Eq. (61))} \\
&\geq \|\boldsymbol{a}\|^2 \\
&= \left\| \boldsymbol{w}_{(1),e} \right\|^2 + \frac{\langle \mathbf{DP}\boldsymbol{w}_{(1),e}, \boldsymbol{w}_{(1),e} \rangle^2}{\left\| \boldsymbol{w}_{(1),e} \right\|^2} - 2\langle \mathbf{DP}\boldsymbol{w}_{(1),e}, \boldsymbol{w}_{(1),e} \rangle \\
&\geq \left\| \boldsymbol{w}_{(1),e} \right\|^2 - 2\langle \mathbf{DP}\boldsymbol{w}_{(1),e}, \boldsymbol{w}_{(1),e} \rangle.
\end{aligned}$$
(62)

We also have

$$\begin{aligned}
\langle \mathbf{DP}\boldsymbol{w}_{(1),e}, \boldsymbol{w}_{(1),e} \rangle &= \boldsymbol{w}_{(1),e}^T \mathbf{DP}\boldsymbol{w}_{(1),e} \\
&= \boldsymbol{w}_{(1),e}^T \mathbf{P}\boldsymbol{w}_{(1),e} \text{ (since } \boldsymbol{w}_{(1),e}^T\mathbf{D} = \boldsymbol{w}_{(1),e}^T \text{ by Eq. (55))} \\
&= \boldsymbol{w}_{(1),e}\mathbf{P}^T\mathbf{P}\boldsymbol{w}_{(1),e} \text{ (by Eq. (34))} \\
&= \left\| \mathbf{P}\boldsymbol{w}_{(1),e} \right\|^2.
\end{aligned}$$
(63)

By Eq. (62) and Eq. (63), we further obtain

$$\begin{aligned}
\mathbb{E} \left\| \boldsymbol{w}_{(1),e} - \mathbf{DP}\boldsymbol{w}_{(1),e} \right\|^2 &\geq \left\| \boldsymbol{w}_{(1),e} \right\|^2 - 2\mathbb{E} \left\| \mathbf{P}\boldsymbol{w}_{(1),e} \right\|^2 \\
&= \left( 1 - \frac{2n_{(1)}}{p + p_{(1)}} \right) \left\| \boldsymbol{w}_{(1),e} \right\|^2 \text{ (by Lemma 7)} \\
&= \left( 1 - \frac{2n_{(1)}}{p + p_{(1)}} \right) \left\| \boldsymbol{w}_{(1)} \right\|^2 \text{ (by Eq. (35))}.
\end{aligned}$$

Therefore, we have proven the lower bound in this lemma. In summary, the result of this lemma thus holds. □

### B.6 PROOF OF LEMMA 19

*Proof.* By Eq. (31) and Eq. (36), we have

$$\mathbf{Z}_{(1)}^T\boldsymbol{q}_{(1)} = \mathbf{U}_{(1)}^T\boldsymbol{q}_{(1),e}.$$
(64)

By Eq. (31), we conclude

$$\mathbf{G}^T\mathbf{G} = \mathbf{G}, \quad \mathbf{G}\boldsymbol{q}_{(1),e} = \mathbf{G}^T\boldsymbol{q}_{(1),e} = \boldsymbol{q}_{(1),e}.$$
(65)

By Eq. (65) and Eq. (34), we obtain

$$\boldsymbol{q}_{(1),e}^T\mathbf{P}^T\mathbf{G}^T\mathbf{G}\mathbf{P}\boldsymbol{q}_{(1),e} = \boldsymbol{q}_{(1),e}^T\mathbf{P}^T\mathbf{G}\mathbf{P}^T\boldsymbol{q}_{(1),e}, \quad \boldsymbol{q}_{(1),e}^T\mathbf{G}\mathbf{P}\boldsymbol{q}_{(1),e} = \boldsymbol{q}_{(1),e}^T\mathbf{P}\boldsymbol{q}_{(1),e}.$$
(66)

By Eq. (34), we have

$$\boldsymbol{q}_{(1),e}^T\mathbf{P}^T\mathbf{P}\boldsymbol{q}_{(1),e} = \boldsymbol{q}_{(1),e}^T\mathbf{P}\boldsymbol{q}_{(1),e}.$$
(67)

Define

$$\boldsymbol{a} := \mathbf{P}\boldsymbol{q}_{(1),e} - \frac{\langle \mathbf{P}\boldsymbol{q}_{(1),e}, \boldsymbol{q}_{(1),e} \rangle}{\left\| \boldsymbol{q}_{(1),e} \right\|^2}\boldsymbol{q}_{(1),e}, \text{ (corresponds to } \overrightarrow{\mathrm{EB}} \text{ in Fig. 2)}$$
(68)

$$\boldsymbol{b} := \mathbf{P}\boldsymbol{q}_{(1),e} - \mathbf{G}\mathbf{P}\boldsymbol{q}_{(1),e} \text{ (corresponds to } \overrightarrow{\mathrm{DB}} \text{ in Fig. 2)}.$$
(69)

Notice that

$$(\boldsymbol{a} - \boldsymbol{b})^T \boldsymbol{b}$$

$$= \boldsymbol{q}_{(1),e}^T \mathbf{P}^T \mathbf{P} \boldsymbol{q}_{(1),e} - \boldsymbol{q}_{(1),e}^T \mathbf{P}^T \mathbf{G} \mathbf{P} \boldsymbol{q}_{(1),e} - \frac{\langle \mathbf{P} \boldsymbol{q}_{(1),e}, \, \boldsymbol{q}_{(1),e} \rangle}{\left\| \boldsymbol{q}_{(1),e} \right\|^2} \boldsymbol{q}_{(1),e}^T \mathbf{P} \boldsymbol{q}_{(1),e}$$

$$+ \frac{\langle \mathbf{P} \boldsymbol{q}_{(1),e}, \, \boldsymbol{q}_{(1),e} \rangle}{\left\| \boldsymbol{q}_{(1),e} \right\|^2} \boldsymbol{q}_{(1),e}^T \mathbf{G} \mathbf{P} \boldsymbol{q}_{(1),e}$$

$$+ \boldsymbol{q}_{(1),e}^T \mathbf{P}^T \mathbf{P} \boldsymbol{q}_{(1),e} - 2 \boldsymbol{q}_{(1),e}^T \mathbf{G} \mathbf{P} \boldsymbol{q}_{(1),e} + \boldsymbol{q}_{(1),e}^T \mathbf{P}^T \mathbf{G}^T \mathbf{G} \mathbf{P} \boldsymbol{q}_{(1),e} \text{ (by Eq. (68) and Eq. (69))}$$

$$= 0 \text{ (by Eq. (66) and Eq. (67)).} \tag{70}$$

Thus, we obtain

$$\left\| C \right\|^2$$

$$= \left\| \mathbf{D} \mathbf{P} \boldsymbol{q}_{(1),e} \right\|^2 \text{ (by Eq. (33) and Eq. (64))}$$

$$= \left\| \mathbf{P} \boldsymbol{q}_{(1),e} - \mathbf{G} \mathbf{P} \boldsymbol{q}_{(1),e} \right\|^2 \text{ (since } \mathbf{G} = \mathbf{I}_{p+p_{(1)}} - \mathbf{D} \text{ by Eq. (31))}$$

$$= \left\| \boldsymbol{b} \right\|^2 \text{ (by Eq. (69))}$$

$$\le \left\| \boldsymbol{a} - \boldsymbol{b} \right\|^2 + \left\| \boldsymbol{b} \right\|^2$$

$$= \left\| \boldsymbol{a} - \boldsymbol{b} \right\|^2 + \left\| \boldsymbol{b} \right\|^2 + 2(\boldsymbol{a} - \boldsymbol{b})^T \boldsymbol{b} \text{ (by Eq. (70))}$$

$$= \left\| \boldsymbol{a} - \boldsymbol{b} + \boldsymbol{b} \right\|^2$$

$$= \left\| \boldsymbol{a} \right\|^2. \tag{71}$$

It remains to estimate the norm of $\boldsymbol{a}$. To that end, we have

$$\left\| \boldsymbol{a} \right\|^2 = \left\| \mathbf{P} \boldsymbol{q}_{(1),e} - \frac{\boldsymbol{q}_{(1),e}^T \mathbf{P} \boldsymbol{q}_{(1),e}}{\left\| \boldsymbol{q}_{(1),e} \right\|^2} \boldsymbol{q}_{(1),e} \right\|^2 \text{ (by Eq. (68))}$$

$$= \left\| \mathbf{P} \boldsymbol{q}_{(1),e} \right\|^2 - \frac{\left( \boldsymbol{q}_{(1),e}^T \mathbf{P} \boldsymbol{q}_{(1),e} \right)^2}{\left\| \boldsymbol{q}_{(1),e} \right\|^2} \text{ (expand and merge like terms)}$$

$$= \left\| \mathbf{P} \boldsymbol{q}_{(1),e} \right\|^2 - \frac{\left\| \mathbf{P} \boldsymbol{q}_{(1),e} \right\|^4}{\left\| \boldsymbol{q}_{(1),e} \right\|^2} \text{ (by Eq. (67))}$$

$$= \left\| \boldsymbol{q}_{(1),e} \right\|^2 \cdot \frac{\left\| \mathbf{P} \boldsymbol{q}_{(1),e} \right\|^2}{\left\| \boldsymbol{q}_{(1),e} \right\|^2} \cdot \left( 1 - \frac{\left\| \mathbf{P} \boldsymbol{q}_{(1),e} \right\|^2}{\left\| \boldsymbol{q}_{(1),e} \right\|^2} \right). \tag{72}$$

Since $\mathbf{P}$ is a projection by Eq. (34), we know $\left\| \boldsymbol{q}_{(1),e} \right\| \ge \left\| \mathbf{P} \boldsymbol{q}_{(1),e} \right\|$. Then, we obtain

$$\frac{\left\| \mathbf{P} \boldsymbol{q}_{(1),e} \right\|^2}{\left\| \boldsymbol{q}_{(1),e} \right\|^2} \le 1, \quad \left( 1 - \frac{\left\| \mathbf{P} \boldsymbol{q}_{(1),e} \right\|^2}{\left\| \boldsymbol{q}_{(1),e} \right\|^2} \right) \le 1. \tag{73}$$

We thus conclude

$$\left\| \boldsymbol{a} \right\|^2 \le \min \left\{ \frac{\left\| \mathbf{P} \boldsymbol{q}_{(1),e} \right\|^2}{\left\| \boldsymbol{q}_{(1),e} \right\|^2}, \, 1 - \frac{\left\| \mathbf{P} \boldsymbol{q}_{(1),e} \right\|^2}{\left\| \boldsymbol{q}_{(1),e} \right\|^2} \right\} \left\| \boldsymbol{q}_{(1),e} \right\|^2 \text{ (by Eq. (72) and Eq. (73)).}$$

Therefore, we have

$$\mathbb{E} \left[ \left\| C \right\|^2 \right] \le \mathbb{E} \left\| \boldsymbol{a} \right\|^2 \text{ (by Eq. (71))}$$

$$\le \min \left\{ \frac{\mathbb{E} \left\| \mathbf{P} \boldsymbol{q}_{(1),e} \right\|^2}{\left\| \boldsymbol{q}_{(1),e} \right\|^2}, \, 1 - \frac{\mathbb{E} \left\| \mathbf{P} \boldsymbol{q}_{(1),e} \right\|^2}{\left\| \boldsymbol{q}_{(1),e} \right\|^2} \right\} \left\| \boldsymbol{q}_{(1),e} \right\|^2$$

$$\le \min \left\{ \left( 1 - \frac{n_{(1)}}{p + p_{(1)}} \right), \, \frac{n_{(1)}}{p + p_{(1)}} \right\} \left\| \boldsymbol{q}_{(1),e} \right\|^2 \text{ (by Lemma 7)}$$

$$= \min \left\{ \left( 1 - \frac{n_{(1)}}{p + p_{(1)}} \right), \, \frac{n_{(1)}}{p + p_{(1)}} \right\} \left\| \boldsymbol{q}_{(1)} \right\|^2 \text{ (by Eq. (36)).}$$

The result of this lemma thus follows. □

## C  PROOF OF THEOREM 2

*Proof.* When $p_{(2)} > n_{(2)} + 1$, we have

$$
\begin{aligned}
\tilde{\boldsymbol{q}}_{(2)} =&\mathbf{Z}_{(2)}(\mathbf{Z}_{(2)}^T\mathbf{Z}_{(2)})^{-1}\left(\boldsymbol{y}_{(2)} - \mathbf{X}_{(2)}^T\tilde{\boldsymbol{w}}_{(1)}\right) \\
=&\mathbf{Z}_{(2)}(\mathbf{Z}_{(2)}^T\mathbf{Z}_{(2)})^{-1}\left(\mathbf{X}_{(2)}^T\left(\boldsymbol{w}_{(2)} - \tilde{\boldsymbol{w}}_{(1)}\right) + \mathbf{Z}_{(2)}^T\boldsymbol{q}_{(2)} + \boldsymbol{\epsilon}_{(2)}\right).
\end{aligned}
$$

Define

$$
\boldsymbol{a} := \left(\mathbf{I}_{p_{(2)}} - \mathbf{Z}_{(2)}(\mathbf{Z}_{(2)}^T\mathbf{Z}_{(2)})^{-1}\mathbf{Z}_{(2)}^T\right)\boldsymbol{q}_{(2)} \in \mathbb{R}^{p_{(2)}}, \tag{74}
$$

$$
\boldsymbol{b} := \left(\mathbf{Z}_{(2)}(\mathbf{Z}_{(2)}^T\mathbf{Z}_{(2)})^{-1}\left(\mathbf{X}_{(2)}^T\left(\boldsymbol{w}_{(2)} - \tilde{\boldsymbol{w}}_{(1)}\right) + \boldsymbol{\epsilon}_{(2)}\right)\right) \in \mathbb{R}^{p_{(2)}}. \tag{75}
$$

Thus, we obtain

$$
\boldsymbol{q}_{(2)} - \tilde{\boldsymbol{q}}_{(2)} = \boldsymbol{a} + \boldsymbol{b}. \tag{76}
$$

By Assumption 1, we conclude

$$
\mathbf{X}_{(2)}^T(\boldsymbol{w}_{(2)} - \tilde{\boldsymbol{w}}_{(1)}) + \boldsymbol{\epsilon}_{(2)} \sim \mathcal{N}\left(\mathbf{0}, \left(\left\|\boldsymbol{w}_{(2)} - \tilde{\boldsymbol{w}}_{(1)}\right\|^2 + \sigma_{(2)}^2\right)\mathbf{I}_{n_{(2)}}\right). \tag{77}
$$

Thus, we have

$$
\begin{aligned}
\underset{\mathbf{X}_{(2)},\boldsymbol{\epsilon}_{(2)}}{\mathbb{E}}[\boldsymbol{b}] =&\underset{\mathbf{X}_{(2)},\boldsymbol{\epsilon}_{(2)}}{\mathbb{E}}\left[\left(\mathbf{Z}_{(2)}(\mathbf{Z}_{(2)}^T\mathbf{Z}_{(2)})^{-1}\left(\mathbf{X}_{(2)}^T\left(\boldsymbol{w}_{(2)} - \tilde{\boldsymbol{w}}_{(1)}\right) + \boldsymbol{\epsilon}_{(2)}\right)\right)\right] \text{ (by Eq. 75)} \\
=&\mathbf{Z}_{(2)}(\mathbf{Z}_{(2)}^T\mathbf{Z}_{(2)})^{-1}\underset{\mathbf{X}_{(2)},\boldsymbol{\epsilon}_{(2)}}{\mathbb{E}}\left[\left(\left(\mathbf{X}_{(2)}^T\left(\boldsymbol{w}_{(2)} - \tilde{\boldsymbol{w}}_{(1)}\right) + \boldsymbol{\epsilon}_{(2)}\right)\right)\right]
\end{aligned}
$$

(since $\mathbf{X}_{(2)}$, $\mathbf{Z}_{(2)}$, and $\boldsymbol{\epsilon}_{(2)}$ are independent by Assumption 1)

$$
=\mathbf{0} \text{ (by Eq. (77))}.
$$

We therefore obtain

$$
\underset{\mathbf{Z}_{(2)},\mathbf{X}_{(2)},\boldsymbol{\epsilon}_{(2)}}{\mathbb{E}}[\boldsymbol{a}^T\boldsymbol{b}] = \underset{\mathbf{Z}_{(2)}}{\mathbb{E}}\left[\boldsymbol{a}^T\underset{\mathbf{X}_{(2)},\boldsymbol{\epsilon}_{(2)}}{\mathbb{E}}[\boldsymbol{b}]\right] = 0. \tag{78}
$$

By Eq. (76) and Eq. (78), we have

$$
\begin{aligned}
&\mathbb{E}\left\|\boldsymbol{q}_{(2)} - \tilde{\boldsymbol{q}}_{(2)}\right\|^2 \\
=&\mathbb{E}\left\|\boldsymbol{a}\right\|^2 + \mathbb{E}\left\|\boldsymbol{b}\right\|^2 \\
=&\left(1 - \frac{n_{(2)}}{p_{(2)}}\right)\left\|\boldsymbol{q}_{(2)}\right\|^2 + \mathbb{E}\left\|\boldsymbol{b}\right\|^2 \text{ (by Eq. (74) and Lemma 7)} \\
=&\left(1 - \frac{n_{(2)}}{p_{(2)}}\right)\left\|\boldsymbol{q}_{(2)}\right\|^2 + \frac{n_{(2)}\left(\left\|\boldsymbol{w}_{(2)} - \tilde{\boldsymbol{w}}_{(1)}\right\|^2 + \sigma_{(2)}^2\right)}{p_{(2)} - n_{(2)} - 1} \text{ (by Eq. (77) and Lemma 8)} \tag{79}
\end{aligned}
$$

When $p_{(2)} + 1 < n_{(2)}$, we obtain

$$
\begin{aligned}
\tilde{\boldsymbol{q}}_{(2)} =&\left(\mathbf{Z}_{(2)}\mathbf{Z}_{(2)}^T\right)^{-1}\mathbf{Z}_{(2)}\left(\boldsymbol{y}_{(2)} - \mathbf{X}_{(2)}^T\tilde{\boldsymbol{w}}_{(1)}\right) \\
=&\left(\mathbf{Z}_{(2)}\mathbf{Z}_{(2)}^T\right)^{-1}\mathbf{Z}_{(2)}\left(\mathbf{X}_{(2)}^T\left(\boldsymbol{w}_{(2)} - \tilde{\boldsymbol{w}}_{(1)}\right) + \mathbf{Z}_{(2)}^T\boldsymbol{q}_{(2)} + \boldsymbol{\epsilon}_{(2)}\right) \\
=&\boldsymbol{q}_{(2)} + \left(\mathbf{Z}_{(2)}\mathbf{Z}_{(2)}^T\right)^{-1}\mathbf{Z}_{(2)}\left(\mathbf{X}_{(2)}^T\left(\boldsymbol{w}_{(2)} - \tilde{\boldsymbol{w}}_{(1)}\right) + \boldsymbol{\epsilon}_{(2)}\right),
\end{aligned}
$$

and therefore we have

$$
\begin{aligned}
\mathbb{E}\left\|\boldsymbol{q}_{(2)} - \tilde{\boldsymbol{q}}_{(2)}\right\|^2 =&\mathbb{E}\left\|\left(\mathbf{Z}_{(2)}\mathbf{Z}_{(2)}^T\right)^{-1}\mathbf{Z}_{(2)}\left(\mathbf{X}_{(2)}^T\left(\boldsymbol{w}_{(2)} - \tilde{\boldsymbol{w}}_{(1)}\right) + \boldsymbol{\epsilon}_{(2)}\right)\right\|^2 \\
=&\frac{p_{(2)}\left(\left\|\boldsymbol{w}_{(2)} - \tilde{\boldsymbol{w}}_{(1)}\right\|^2 + \sigma_{(2)}^2\right)}{n_{(2)} - p_{(2)} - 1} \text{ (by Eq. (77) and Lemma 8)}. \tag{80}
\end{aligned}
$$

By Eq. (79) and Eq. (80), the result of this proposition thus follows. □

# D PROOF OF THEOREM 3

Define

$$\mathbf{U}_{(2)} := \begin{bmatrix} \mathbf{X}_{(2)} \\ \mathbf{Z}_{(2)} \end{bmatrix} \in \mathbb{R}^{(p+p_{(2)}) \times n_{(2)}}.$$

## D.1 OVERFITTED SITUATION

We first consider the overfitted solution, i.e., $p+p_{(2)} > n_{(2)}$. The overfitted solution that corresponds to the convergence of GD/SGD is given by

$$\min_{\hat{\boldsymbol{w}}_{(2)}, \hat{\boldsymbol{q}}_{(2)}} \quad \left\| \hat{\boldsymbol{w}}_{(2)} - \tilde{\boldsymbol{w}}_{(1)} \right\|^2 + \left\| \hat{\boldsymbol{q}}_{(2)} \right\|^2 \tag{81}$$
$$\text{subject to} \quad \boldsymbol{y}_{(2)} = \mathbf{X}_{(2)}^T \hat{\boldsymbol{w}}_{(2)} + \mathbf{Z}_{(2)}^T \hat{\boldsymbol{q}}_{(2)}.$$

The solution of (81) is as follows:

$$\begin{aligned}
\begin{bmatrix} \tilde{\boldsymbol{w}}_{(2)} - \tilde{\boldsymbol{w}}_{(1)} \\ \hat{\boldsymbol{q}}_{(2)} \end{bmatrix} &= \mathbf{U}_{(2)} (\mathbf{U}_{(2)}^T \mathbf{U}_{(2)})^{-1} \left( \boldsymbol{y}_{(2)} - \mathbf{X}_{(2)}^T \tilde{\boldsymbol{w}}_{(1)} \right) \\
&= \mathbf{U}_{(2)} (\mathbf{U}_{(2)}^T \mathbf{U}_{(2)})^{-1} \left( \mathbf{X}_{(2)}^T (\boldsymbol{w}_{(2)} - \tilde{\boldsymbol{w}}_{(1)}) + \mathbf{Z}_{(2)}^T \boldsymbol{q}_{(2)} + \boldsymbol{\epsilon}_{(2)} \right) \quad \text{(by Eq. (3))} \\
&= \mathbf{U}_{(2)} (\mathbf{U}_{(2)}^T \mathbf{U}_{(2)})^{-1} \boldsymbol{\epsilon}_{(2)} + \mathbf{U}_{(2)} (\mathbf{U}_{(2)}^T \mathbf{U}_{(2)})^{-1} \mathbf{U}_{(2)}^T \begin{bmatrix} \boldsymbol{w}_{(2)} - \tilde{\boldsymbol{w}}_{(1)} \\ \boldsymbol{q}_{(2)} \end{bmatrix}. \tag{82}
\end{aligned}$$

By Assumption 1, we know that $\boldsymbol{\epsilon}_{(2)}$ has zero mean and is independent of $\mathbf{U}_{(2)}$. Thus, we have

$$\left\langle \left( \mathbf{I}_{p+p_{(2)}} - \mathbf{U}_{(2)} (\mathbf{U}_{(2)}^T \mathbf{U}_{(2)})^{-1} \mathbf{U}_{(2)}^T \right) \begin{bmatrix} \boldsymbol{w}_{(2)} - \tilde{\boldsymbol{w}}_{(1)} \\ \boldsymbol{q}_{(2)} \end{bmatrix}, \ \mathbf{U}_{(2)} (\mathbf{U}_{(2)}^T \mathbf{U}_{(2)})^{-1} \boldsymbol{\epsilon}_{(2)} \right\rangle = 0. \tag{83}$$

We therefore obtain

$$\begin{aligned}
& \mathbb{E} \left[ \left\| \boldsymbol{w}_{(2)} - \tilde{\boldsymbol{w}}_{(2)} \right\|^2 + \left\| \boldsymbol{q}_{(2)} - \tilde{\boldsymbol{q}}_{(2)} \right\|^2 \right] \\
=& \mathbb{E} \left\| \begin{bmatrix} \boldsymbol{w}_{(2)} - \tilde{\boldsymbol{w}}_{(1)} \\ \boldsymbol{q}_{(2)} \end{bmatrix} - \begin{bmatrix} \tilde{\boldsymbol{w}}_{(2)} - \tilde{\boldsymbol{w}}_{(1)} \\ \tilde{\boldsymbol{q}}_{(2)} \end{bmatrix} \right\|^2 \\
=& \mathbb{E} \left\| \left( \mathbf{I}_{p+p_{(2)}} - \mathbf{U}_{(2)} (\mathbf{U}_{(2)}^T \mathbf{U}_{(2)})^{-1} \mathbf{U}_{(2)}^T \right) \begin{bmatrix} \boldsymbol{w}_{(2)} - \tilde{\boldsymbol{w}}_{(1)} \\ \boldsymbol{q}_{(2)} \end{bmatrix} + \mathbf{U}_{(2)} (\mathbf{U}_{(2)}^T \mathbf{U}_{(2)})^{-1} \boldsymbol{\epsilon}_{(2)} \right\|^2 \quad \text{(by Eq. (82))} \\
=& \mathbb{E} \left\| \left( \mathbf{I}_{p+p_{(2)}} - \mathbf{U}_{(2)} (\mathbf{U}_{(2)}^T \mathbf{U}_{(2)})^{-1} \mathbf{U}_{(2)}^T \right) \begin{bmatrix} \boldsymbol{w}_{(2)} - \tilde{\boldsymbol{w}}_{(1)} \\ \boldsymbol{q}_{(2)} \end{bmatrix} \right\|^2 + \mathbb{E} \left\| \mathbf{U}_{(2)} (\mathbf{U}_{(2)}^T \mathbf{U}_{(2)})^{-1} \boldsymbol{\epsilon}_{(2)} \right\|^2 \quad \text{(by Eq. (83))} \\
=& \left( 1 - \frac{n_{(2)}}{p + p_{(2)}} \right) \left\| \begin{matrix} \boldsymbol{w}_{(2)} - \tilde{\boldsymbol{w}}_{(1)} \\ \boldsymbol{q}_{(2)} \end{matrix} \right\|^2 + \frac{n_{(2)} \sigma_{(2)}^2}{p + p_{(2)} - n_{(2)} - 1} \quad \text{(by Lemma 7 and Lemma 8)} \\
=& \left( 1 - \frac{n_{(2)}}{p + p_{(2)}} \right) \left( \left\| \boldsymbol{w}_{(2)} - \tilde{\boldsymbol{w}}_{(1)} \right\|^2 + \left\| \boldsymbol{q}_{(2)} \right\|^2 \right) + \frac{n_{(2)} \sigma_{(2)}^2}{p + p_{(2)} - n_{(2)} - 1}.
\end{aligned}$$

## D.2 UNDERPARAMETERIZED SITUATION

The solution that minimizes the training loss is given by

$$\begin{aligned}
\begin{bmatrix} \tilde{\boldsymbol{w}}_{(2)} \\ \tilde{\boldsymbol{q}}_{(2)} \end{bmatrix} &= (\mathbf{U}_{(2)} \mathbf{U}_{(2)}^T)^{-1} \mathbf{U}_{(2)} \boldsymbol{y}_{(2)} \\
&= (\mathbf{U}_{(2)} \mathbf{U}_{(2)}^T)^{-1} \mathbf{U}_{(2)} \left( \mathbf{U}_{(2)}^T \begin{bmatrix} \boldsymbol{w}_{(2)} \\ \boldsymbol{q}_{(2)} \end{bmatrix} + \boldsymbol{\epsilon}_{(2)} \right) \quad \text{(by Eq. (3))} \\
&= \begin{bmatrix} \boldsymbol{w}_{(2)} \\ \boldsymbol{q}_{(2)} \end{bmatrix} + (\mathbf{U}_{(2)} \mathbf{U}_{(2)}^T)^{-1} \mathbf{U}_{(2)} \boldsymbol{\epsilon}_{(2)}.
\end{aligned}$$

Thus, we have

$$\begin{aligned}
\mathbb{E} \left[ \left\| \boldsymbol{w}_{(2)} - \tilde{\boldsymbol{w}}_{(2)} \right\|^2 + \left\| \boldsymbol{q}_{(2)} - \tilde{\boldsymbol{q}}_{(2)} \right\|^2 \right] &= \left\| (\mathbf{U}_{(2)} \mathbf{U}_{(2)}^T)^{-1} \mathbf{U}_{(2)} \boldsymbol{\epsilon}_{(2)} \right\|^2 \\
&= \frac{(p + p_{(2)}) \sigma_{(2)}^2}{n_{(2)} - (p + p_{(2)}) - 1} \quad \text{(by Lemma 8)}.
\end{aligned}$$

# E    PROOF OF PROPOSITION 4

*Proof.* We use $[\cdot]_i$ to denote the $i$-th element of a vector. Then we have

$$
\begin{aligned}
&\left\| \boldsymbol{w}_{(2)} - \tilde{\boldsymbol{w}}_{(1)} \right\|^2 \\
&= \sum_{i=1}^{p} \left( [\boldsymbol{w}_{(2)}]_i - [\tilde{\boldsymbol{w}}_{(1)}]_i \right)^2 \quad (\text{since } \boldsymbol{w}_{(2)}, \tilde{\boldsymbol{w}}_{(1)} \in \mathbb{R}^p) \\
&= \sum_{i \in \hat{\mathcal{S}}_{\text{co}}} \left( [\boldsymbol{w}_{(2)}]_i - [\tilde{\boldsymbol{w}}_{(1)}]_i \right)^2 + \sum_{i \in \mathcal{S}_{\text{co}} \setminus \hat{\mathcal{S}}_{\text{co}}} [\tilde{\boldsymbol{w}}_{(1)}]_i^2 \quad (\text{by the definition of } \boldsymbol{w}_{(2)}).
\end{aligned}
\tag{84}
$$

We thus obtain

$$
\begin{aligned}
\mathcal{L}_{\text{co}} &= \mathbb{E} \left\| \boldsymbol{w}_{(2)} - \tilde{\boldsymbol{w}}_{(1)} \right\|^2 \quad (\text{by Eq. (5)}) \\
&= \mathbb{E} \sum_{i \in \hat{\mathcal{S}}_{\text{co}}} \left( [\boldsymbol{w}_{(2)}]_i - [\tilde{\boldsymbol{w}}_{(1)}]_i \right)^2 + \mathbb{E} \sum_{i \in \mathcal{S}_{\text{co}} \setminus \hat{\mathcal{S}}_{\text{co}}} [\tilde{\boldsymbol{w}}_{(1)}]_i^2 \quad (\text{by Eq. (84)}).
\end{aligned}
$$

Since $p + p_{(1)} = C$ is fixed and all features are *i.i.d.* Gaussian (Assumption 1), the distribution of $\begin{bmatrix} \tilde{\boldsymbol{w}}_{(1)} \\ \tilde{\boldsymbol{q}}_{(1)} \end{bmatrix} \in \mathbb{R}^C$ is the same for different $p$. Notice that since Definition 1 is assured, the term $\mathbb{E} \sum_{i \in \hat{\mathcal{S}}_{\text{co}}} \left( [\boldsymbol{w}_{(2)}]_i - [\tilde{\boldsymbol{w}}_{(1)}]_i \right)^2$ does not change with respect to $p$. In contrast, the term $\mathbb{E} \sum_{i \in \mathcal{S}_{\text{co}} \setminus \hat{\mathcal{S}}_{\text{co}}} [\tilde{\boldsymbol{w}}_{(1)}]_i^2$ is monotone increasing with respect to $p$ (since $\mathcal{S}_{\text{co}} \setminus \hat{\mathcal{S}}_{\text{co}}$ includes more fake features when $p$ is larger). The result of Proposition 4 thus follows. $\qquad \square$

# F    ADDITIONAL SIMULATION RESULTS

Fig. 4 shows the results under correlated (instead of *i.i.d.*) Gaussian features where the covariance between any two different standard Gaussian features of a task is 0.5. We find that the shapes of those curves are very similar to the ones in Fig. 1, which implies that most of the insights still hold in the moderate non-i.i.d. setup.

In Fig. 5, we conduct experiments using a 5-layer CNN to do transfer learning on CIFAR-10. By increasing the width of the transferred layers (corresponding to $p$ in our linear setup), we can observe the descent in the overfitted regime. We can also observe that Option B performs better than Option A, which is consistent with our discussion in Section 4.1.

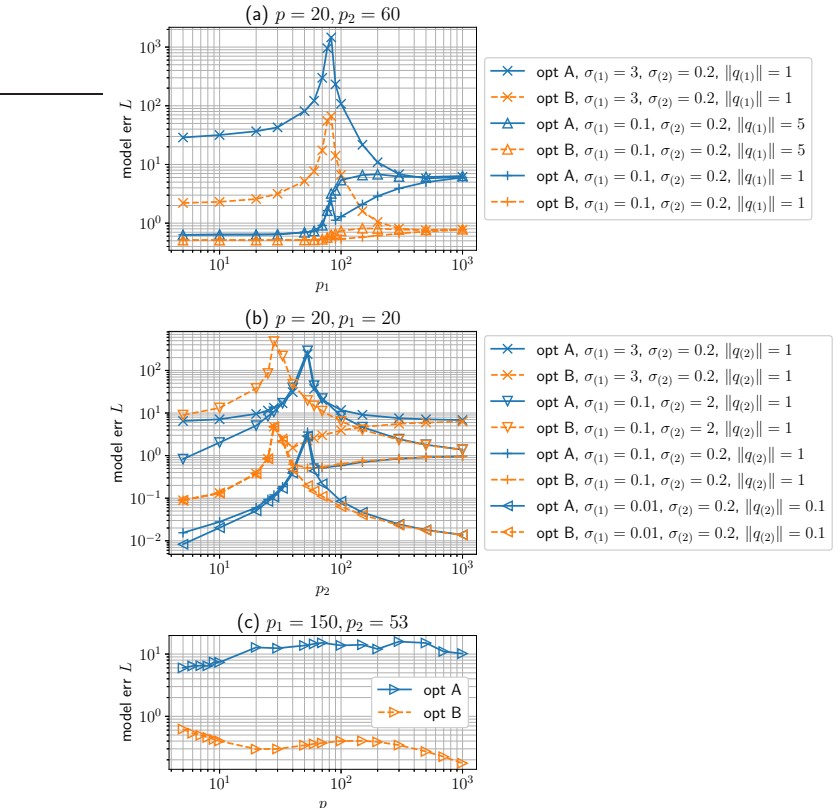

Figure 4: Generalization performance of transfer learning with correlated Gaussian features where the covariance between any two different standard Gaussian features of a task is 0.5. Other settings are the same as those of Fig. 1.

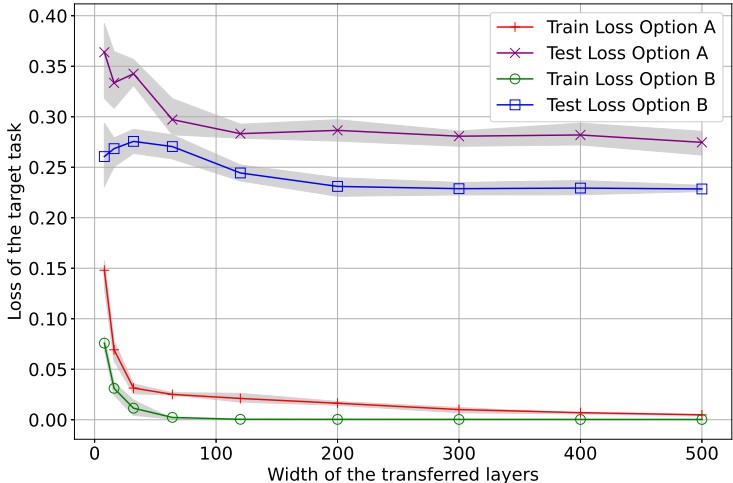

Figure 5: Benign overfitting w.r.t. the width of the transferred layers in a 5-layer CNN. Specifically, a custom CNN (2 convolutional layers followed by 3 fully connected layers) is designed to perform transfer learning on two tasks using the CIFAR-10 dataset. Task 1 classifies images of airplanes and automobiles, while Task 2 classifies ships and trucks. The model's bottom layers (2 convolutional layers) are transferred from Task 1 to Task 2. Option A freezes these transferred layers, while Option B continues to train them. The loss function used in this experiment is the Cross-Entropy Loss. Each point in the figure is the average of 10 random runs.

