# OpenReview forum: "Theoretical Analysis on the Generalization Power of Overfitted Transfer Learning"
_ICLR.cc/2024/Conference — Submitted to ICLR 2024_

### Official Review · Reviewer_onS3 · 2023-10-28

**Soundness:** 3 good
**Presentation:** 2 fair
**Contribution:** 2 fair
**Rating:** 5
**Confidence:** 3

**Summary:**

The paper studies transfer learning under the noisy linear regression setting. The authors split the features into two distinct sets: a common part shared across tasks and a task-specific part. The paper studies two transfer learning settings, Option A: directly copy the learned common feature to the target task; Option B: use the learned common part as an initial training point. Then, considering different noise levels, the authors claim that, when the total number of features in the source task’s learning model is fixed, it is more advantageous to allocate a greater number of redundant features to the task-specific part rather than the common part.

**Strengths:**

The paper studies an important topic. Nowadays, transfer learning is become the new standard paradigm. For example, training a foundation model by self-supervised learning and adapting the model to a specific supervised downstream task. Thus, some important topics include but are not limited to how to select features, how to select the aulixray dataset, how to adapt the models efficiently, and so on.

The paper’s motivation is clear and the analysis is concise.

**Weaknesses:**

- The major concern I have is that the paper can separate the common features and task-specific features explicitly. In other words, in Option A and Option B, we can directly know that $w$ is a common feature, while $q$ is not. In practice, we do not have any idea about which part of the feature should be useful and we should try to figure out the common features and task-specific features rather than assuming we know that. From my perspective, one practical Option C should use all the learned features, e.g., $w$ and $q$  as an initial training point rather than $w$ only in Option B.
- The problem setup is simple, not practical, and not novel. The linear regression setting seems not surprising and the transfer learning conclusion directly depends mostly on the assumption we made. The paper only considers one source supervised training task, while in practice, we mostly consider multiple source tasks [1,2] or different pretraining objectives, e.g., self-supervised learning [3,4] (no objective gap between source and target). Splitting the data feature space into common features and task-specific features is quite commonly used [4,5]. While in [4,5] they do not know which part is a common feature during pretraining and transfer learning.
- The claim in Proposition 4 is not significant. From my perspective, it just says that we should only keep the common features in the transferred feature, where the take-home message is not interesting. I cannot get more insights or intuitions.
- Some minor weaknesses in writing. Assumption 1 should be Definition 1. $\mathcal{R}$ and $\mathbb{R}$ are mixed used.

[1] Nilesh Tripuraneni, Michael Jordan, and Chi Jin. On the theory of transfer learning: The importance of task diversity. NeurIPS 2020.

[2] Zhao, Yulai, Jianshu Chen, and Simon Du. Blessing of Class Diversity in Pre-training. AISTATS 2023.

[3] HaoChen, Jeff Z., Colin Wei, Adrien Gaidon, and Tengyu Ma. Provable guarantees for self-supervised deep learning with spectral contrastive loss.  NeurIPS 2021.

[4] Shi, Zhenmei, Jiefeng Chen, Kunyang Li, Jayaram Raghuram, Xi Wu, Yingyu Liang, and Somesh Jha. The Trade-off between Universality and Label Efficiency of Representations from Contrastive Learning. ICLR 2023.

[5] Rosenfeld, Elan, Pradeep Kumar Ravikumar, and Andrej Risteski. The Risks of Invariant Risk Minimization. ICLR 2021.

**Questions:**

In Remark 1, it is unclear what missing features mean. Providing a toy example will be good.

---

> ### Author Response · Authors · 2023-11-22
> **Response to Reviewer onS3 (part 1)**
>
> > The major concern I have is that the paper can separate the common features and task-specific features explicitly. In other words, in Option A and Option B, we can directly know that $w$ is a common feature, while $q$ is not. In practice, we do not have any idea about which part of the feature should be useful and we should try to figure out the common features and task-specific features rather than assuming we know that. From my perspective, one practical Option C should use all the learned features, e.g., $w$ and $q$ as an initial training point rather than $w$ only in Option B.
>
> **Response**: We thank the reviewer for raising this important concern. For several modern ML applications, knowing features in advance is very common via pre-training over a big collection of tasks (generally referred to as feature engineering), so that these features can benefit later downstream tasks without the need to reconstruct these features again in each individual task. In particular, recent huge successes such as contrast learning and pre-training large language models (LLMs) can provide powerful pre-trained and general-purpose features/representations (e.g., [ref a], [ref b]). These new ML advances motivate/justify the line of research on transfer learning (which our paper follows) to leverage those well-trained features beforehand and focus on designing the transfer of other model parameters to benefit target tasks. In addition to [ref a] and [ref b], the idea of incorporating **known** features into the learning formulation has become popular in other ML studies (such as in few-shot learning [ref b]) and has been well accepted in the ML community. We will improve our discussion of these points in the paper to address this important concern raised by the reviewer.
>
> [a] Kaiming He, et. al, “Momentum Contrast for Unsupervised Visual Representation Learning” CVPR 202.
>
> [b] Tom Brown et. al, “Language Models are Few-Shot Learners” NeurIPS 2020.
>
> ---
>
> > The problem setup is simple, not practical, and not novel. The linear regression setting seems not surprising and the transfer learning conclusion directly depends mostly on the assumption we made. The paper only considers one source-supervised training task, while in practice, we mostly consider multiple source tasks [1,2] or different pretraining objectives, e.g., self-supervised learning [3,4] (no objective gap between source and target). Splitting the data feature space into common features and task-specific features is quite commonly used [4,5]. While in [4,5] they do not know which part is a common feature during pretraining and transfer learning.
>
> > [1] Nilesh Tripuraneni, Michael Jordan, and Chi Jin. On the theory of transfer learning: The importance of task diversity. NeurIPS 2020.
>
> > [2] Zhao, Yulai, Jianshu Chen, and Simon Du. Blessing of Class Diversity in Pre-training. AISTATS 2023.
>
> > [3] HaoChen, Jeff Z., Colin Wei, Adrien Gaidon, and Tengyu Ma. Provable guarantees for self-supervised deep learning with spectral contrastive loss. NeurIPS 2021.
>
> > [4] Shi, Zhenmei, Jiefeng Chen, Kunyang Li, Jayaram Raghuram, Xi Wu, Yingyu Liang, and Somesh Jha. The Trade-off between Universality and Label Efficiency of Representations from Contrastive Learning. ICLR 2023.
>
> > [5] Rosenfeld, Elan, Pradeep Kumar Ravikumar, and Andrej Risteski. The Risks of Invariant Risk Minimization. ICLR 2021.
>
> **Response**: We thank the reviewer for raising these important concerns. However, using a linear model and only one source task as an initial step is common in the field of theoretical transfer learning in order to glean important insights. Note that although the model is simplified, the analysis is already quite complex and provides interesting insights. Possible future work using more realistic models will also benefit from our work with the simple model. Gaining an understanding of the simple model is fundamentally important to gaining an understanding of more realistic models.
>
> The source task in [4] pre-trains a representation using unlabeled data, while ours uses labeled data. This is exactly why we need to know features but [4] does not (since the goal of [4] is to learn representations but ours is to learn the parameters). [5] focuses on out-of-distribution generalization and is not related to transfer learning.
>
> ---
>
> **Followed by "Response to Reviewer onS3 (part 2)"**

---

> > ### Author Response · Authors · 2023-11-22
> > **Response to Reviewer onS3 (part 2)**
> >
> > **Continued from "Response to Reviewer onS3 (part 1)"**
> >
> > ---
> >
> > > The claim in Proposition 4 is not significant. From my perspective, it just says that we should only keep the common features in the transferred feature, where the take-home message is not interesting. I cannot get more insights or intuitions.
> >
> > **Response**: Thank you for this comment. We agree that Prop. 4 is somewhat intuitive, but it provides important preparation for the surprising message just below Prop. 4 that sometimes it is even better to sacrifice certain true features in the common part. In other words, some true common features can be discarded to gain better performance, which is counter-intuitive. We will clarify this important point in the paper.
> >
> > ---
> >
> > > Some minor weaknesses in writing. Assumption 1 should be Definition 1. R and R are mixed used.
> >
> > **Response**: Thanks for your valuable suggestions. We have modified those places accordingly in the revision.
> >
> > ---
> >
> > > In Remark 1, it is unclear what missing features mean. Providing a toy example will be good.
> >
> > **Response**: Thank you for this comment. A missing feature means that a true feature is not included in the data. For example, to predict the position of a car, true features include velocity, direction, time, acceleration, etc. If in the dataset the direction is not included, then the direction becomes a missing feature. We clarified this in the revision of the paper.
> >
> > ---

---

> > > ### Comment · Reviewer_onS3 · 2023-11-22
> > >
> > > Thank you for your response. My first two concerns are not fully solved and I tend to keep my original score.

---

### Official Review · Reviewer_6fkR · 2023-11-01

**Soundness:** 4 excellent
**Presentation:** 4 excellent
**Contribution:** 4 excellent
**Rating:** 8
**Confidence:** 3

**Summary:**

This paper presents a theoretical analysis of transfer learning in the under-parameterized and over-parameterized scenarios. The paper explores two options for transfer: 1) option A, where the common parameters are copied and only the task specific parameters are trained for the target task, 2) option B, where the common parameters are initialized for the target task and both the common and task specific parameters are trained during that task. The paper presents theoretical results, along with some experiments, for the parameter transfer setting, such as when benign overfitting occurs, what are the effects of noise, and the comparison of the common vs. task-specific parameters for both options.

**Strengths:**

- The paper presents a much needed theoretical analysis of some interesting scenarios in parameter transfer. The contributions seem original and of significance for researchers in transfer learning and related areas.
- The paper is very well-written and structured.
- The paper presents the ideas in a clear way for a theoretical paper. The concepts are presented in a structured manner, and some experimental results are provided to support the theoretical claims.

**Weaknesses:**

- Understandability of some key parts of the paper could be improved. For example, for "Option A" and "Option B", considering how relevant these are for the main results of the paper, perhaps more meaningful names could have been used so it is easy to grasp and associate theoretical results with what's going on in these of these "options".

**Questions:**

- Is there any connection between your work and well-known theoretical studies in multitask learning? [1] Or is your analysis also applicable to multitask learning scenarios? There may be a strong connection since a lot of approaches in multitask learning also rely on the common/task-specific features paradigm.

[1] Baxter, J. (2000). A model of inductive bias learning. Journal of artificial intelligence research, 12, 149-198.

---

> ### Author Response · Authors · 2023-11-22
>
> > Understandability of some key parts of the paper could be improved. For example, for "Option A" and "Option B", considering how relevant these are for the main results of the paper, perhaps more meaningful names could have been used so it is easy to grasp and associate theoretical results with what's going on in these of these "options".
>
> **Response**: We thank the reviewer for these helpful suggestions. In light of the reviewer’s suggestion, in Sec. 2.4 of the revision, we have now renamed the options as Option A – “Transfer and Fix”, Option B – “Transfer and Train”.
>
> ---
>
> > Is there any connection between your work and well-known theoretical studies in multitask learning? [1] Or is your analysis also applicable to multitask learning scenarios? There may be a strong connection since a lot of approaches in multitask learning also rely on the common/task-specific features paradigm.
>
> > [1] Baxter, J. (2000). A model of inductive bias learning. Journal of artificial intelligence research, 12, 149-198.
>
> **Response**: We thank the reviewer for these important questions and suggestions. To the best of our knowledge, there is no multitask learning result similar to ours that studies the overfitted generalization performance with common/task-specific features separation.
>
> In [1] and an extensive line of research on multi-task learning, the focus was on the **under-paraemeterized** regime, where the sample size should be large enough (larger than the complexity of the model class in terms of, e.g., VC-dimension or covering number). In contrast, our focus is on the **over-parameterized** regime, where we characterize that the desirable test performance is still achievable, i.e., benign overfitting can occur.
>
> If the reviewer has any specific work in mind, we are more than happy to cite and make comparisons with this work. We also plan to investigate the case of multiple source/target tasks as future work, building on and exploiting our current analysis.
>
> ---

---

### Official Review · Reviewer_hznS · 2023-11-01

**Soundness:** 2 fair
**Presentation:** 1 poor
**Contribution:** 1 poor
**Rating:** 3
**Confidence:** 4

**Summary:**

This paper studies the generalization of transfer learning in linear regression models under both underparameterized and overparameterized regimes. This work proposes to partition the feature space into common and task-specific parts and analyze the influence of the number of parameters on the generalization performance. This paper presents the transferring errors for two types of transfer learning: the first one fixes the common features and the second one further trains the model initializes from the parameters corresponding to the common features. This paper provides two insights: First, this paper shows that allocating more features to task-specific parts benefits the target task more than allocating to the common part. Second, This paper finds that under high noise and small parameter regimes, sacrificing certain features in the common part and adding more to the task-specific part can yield better generalization.

**Strengths:**

This work proposes to partition the feature space into common and task-specific parts and analyze the influence of the number of parameters on the generalization performance. The results provide insights into choosing different transferring methods under linear regression settings.

**Weaknesses:**

- It is unclear what the contributions this paper makes to the existing literature. It would be better to specify which existing works are compared to when discussing them in the introduction and the main text.
- The terms used in this paper need further specification. For example, it would be better to provide formal definitions of generalization performance, transferring errors, benign overfitting, and descent floors.

**Questions:**

- What does the second step ("extract and transfer the learned parameters of the common features")  of parameter transfer mean? It would be better to elaborate on this step.
- It would be better to provide a formal definition of the transferring error presented in Theorem 1.
- Can the insights from the presented theorems be extended to transfer learning with deep neural networks? Although the theorems are presented for linear regression settings, it would be better to discuss the potential implications for deep neural networks.

---

> ### Author Response · Authors · 2023-11-22
> **Response to Reviewer hznS (part 1)**
>
> We thank the reviewer for their valuable and constructive comments.
>
> > It is unclear what the contributions this paper makes to the existing literature. It would be better to specify which existing works are compared to when discussing them in the introduction and the main text.
>
> **Response**: We thank the reviewer for these valuable comments. Our contribution is as follows:  In this paper, we investigate the generalization performance of transfer learning in linear regression models under both the underparameterized and overparameterized regimes.
> Compared to the existing literature that considers a general noisy linear relation between the true parameters of the source and target tasks, we delve into the separation between common and task-specific features in greater detail.
> Specifically, we partition the feature space into a common part and a task-specific part. This setup enables us to analyze how the number of parameters in different parts influences the generalization performance of the target task. By characterizing the generalization performance, we offer insightful findings on transfer learning. For instance, when the total number of features in the source task's learning model is fixed, our analysis reveals the advantage of \emph{allocating more redundant features to the task-specific part rather than the common part}. Additionally, in specific scenarios characterized by high noise levels and small true parameters, *sacrificing certain true features in the common part in favor of employing more redundant features in the task-specific part can yield notable benefits*.
>
> The most related work to ours is [ref 1]. Specifically, [ref 1] studies the double descent phenomenon in transfer learning, which is also our focus in this paper. However, [ref 1] does not consider an explicit separation of the feature space by the common part and the task-specific part like we do in this paper. As we show, such a separation in the system model enables us to analyze the double descent phenomenon under different options for transfer learning, including two options for parameter transfer and two options for data transfer. In contrast, [ref 1] only studies one option of parameter transfer. Therefore, our analysis is quite different from that of [ref 1].
>
> While the contributions of this paper are summarized in the last paragraph of the Introduction and the comparison to the existing works are introduced in Sec. 1.1, we will improve our discussion in the paper to clarify these important points.
>
> [1] Yehuda Dar and Richard G Baraniuk. Double double descent: on generalization errors in transfer learning between linear regression tasks. SIAM Journal on Mathematics of Data Science, 4(4): 1447–1472, 2022.
>
> ---
>
> > The terms used in this paper need further specification. For example, it would be better to provide formal definitions of generalization performance, transferring errors, benign overfitting, and descent floors.
>
> **Response**: Thank you for these valuable suggestions. Please note that the formal definitions of performance evaluation and transferring errors are given in Sec. 2.5. “Benign overfitting” means that the test error of the overparameterized regime is lower than that of the underparameterized regime. “Descent floor” means that (in the overparameterized regime) the descent of the test error stops at a certain point (which is like a floor). We have revised the presentation in the paper to make these points clear.
>
> ---
>
> > What does the second step ("extract and transfer the learned parameters of the common features") of parameter transfer mean? It would be better to elaborate on this step.
>
> **Response**: We thank the reviewer for this question and suggestion. The second step means selecting the parameters for the common features $S_{co}$ from the learned result of the source task and then sending them to the target task model. We have improved our discussion in the paper to clarify this important point.
>
> ---
>
> > It would be better to provide a formal definition of the transferring error presented in Theorem 1.
>
> **Response**: Thank you for this valuable suggestion. The formal definition of the transferring error is given by Eq. (5). We have clarified this important point in the revision.
>
> ---
> **Followed by "Response to Reviewer hznS (part 2)"**

---

> > ### Author Response · Authors · 2023-11-22
> > **Response to Reviewer hznS (part 2)**
> >
> > **Continued from "Response to Reviewer hznS (part 1)"**
> >
> > > Can the insights from the presented theorems be extended to transfer learning with deep neural networks? Although the theorems are presented for linear regression settings, it would be better to discuss the potential implications for deep neural networks.
> >
> > **Response**: We thank the reviewer for this important question and suggestion. The answer is “yes”. In Fig. 5 in Appendix F, we conduct experiments using a 5-layer CNN to do transfer learning on CIFAR-10. By increasing the width of the transferred layers (corresponding to p in our linear setup), we can observe the descent in the overfitted regime. We can also observe that Option B performs better than Option A, which is consistent with our discussion in Section 4.1. We will revise the paper to elaborate on the potential implications of our work for deep neural networks.
> >
> > ---

---

### Official Review · Reviewer_kvri · 2023-11-03

**Soundness:** 3 good
**Presentation:** 2 fair
**Contribution:** 2 fair
**Rating:** 3
**Confidence:** 4

**Summary:**

The paper explores transfer learning in a linear regression model, wherein certain features are shared between the source and target, while other features are specific to each. Then it analyzes two transfer options:
 A) The learner first learns the source parameter and transfers the parameters corresponding to the shared feature to the target. Afterward, it learns only the parameters corresponding to the target-specific features.
 B) The learner first learns the source parameters and then uses the learned source parameters as an initialization for the target parameters corresponding to the shared features.

Then, by considering both overparameterized and underparameterized settings, it analyzes the behavior of target generalization error as the number of features and parameters varies.

**Strengths:**

The paper analyzes the target generalization error as a function of the number of common and task-specific features and parameters as well as the variance of the noise in the model. Furthermore, it characterizes the regimes in which benign overfitting happens. For instance, they show that in some cases it would be more beneficial to allocate more redundant features to the task-specific part rather than the shared one.

**Weaknesses:**

The model assumed in the paper is quite restrictive. Firstly, it only considers a linear model. Even more restrictive is the assumption that exact shared features exist between the source and target datasets. Additionally, the paper assumes that the shared features are included in the training examples, and the training examples may contain at most some redundant features. In other words, by eliminating only some of the features, one can recover both the shared features and task-specific features.

Certainly, when the source and target share the same features, they become transferable. However, the reverse is not necessarily true. In a previous study [1], it was demonstrated that in a linear regression model, transferability can still occur even if the datasets lack exact shared features, as long as their ground truth parameters have a small distance according to the Frobenius norm.

Furthermore, the literature on transfer learning contains numerous results where source and target training examples do not initially include shared common features. However, through certain transformations, it is possible to find a subspace in which the source and target datasets share common features.

Moreover, the paper has only derived some formulas for generalization error, and it only explains what would happen if one varies the parameters. It does not provide insight into why varying the parameters in certain regimes results in an increase or decrease in generalization error. The paper does not go beyond explaining the algebraic relationships.


[1] SM Mousavi Kalan, Z Fabian, S Avestimehr, M Soltanolkotabi; Minimax Lower Bounds for Transfer Learning with Linear and One-hidden Layer Neural Networks.

**Questions:**

I recommend the authors generalize their results to the case where initially the source and target training examples do not contain shared and task-specific features. However, there exists an unknown transformation after which one can discover exact shared features.
Alternatively, if some of the ground truth parameters between the source and target are close to each other, though not necessarily the same, it would be interesting to investigate and characterize benign overfitting in these more practical scenarios.

---

> ### Author Response · Authors · 2023-11-22
>
> We thank the reviewer for their valuable and constructive comments.
>
> 1. **Linear model**: Using a linear model as an initial step is common in the field of theoretical study. Possible future work using more realistic models will also benefit from our work with the linear model. Gaining an understanding of the simple model is fundamentally important to gaining an understanding of more realistic models.
>
> 2. **Assumptions of knowing features**: For certain applications of machine learning, knowing features in advance is very common in modern machine learning (ML), via pre-training over a big collection of tasks (generally referred to as feature engineering), so that these features can benefit later downstream tasks without the need of reconstructing these features again in each individual task. In particular, recent huge successes such as contrast learning and pre-training large language models (LLMs) can provide powerful pre-trained and general-purpose features/representations (e.g., [ref a], [ref b]). These new ML advances motivate/justify the line of research on transfer learning (which our paper follows) to leverage those well-trained features beforehand and focus on designing the transfer of other model parameters to benefit target tasks. In addition to the above references, the idea of incorporating **known** features into the learning formulation has become popular in other ML studies (such as in few-shot learning [ref b]) and has been well accepted in the ML community. We will improve our discussion of these points in the paper to address this important concern raised by the reviewer.
>
> [a] Kaiming He, et. al, “Momentum Contrast for Unsupervised Visual Representation Learning” CVPR 202.
>
> [b] Tom Brown et. al, “Language Models are Few-Shot Learners” NeurIPS 2020.
>
>
> 3. **“..the training examples may contain at most some redundant features..”**: We also allow missing features by Remark 1.
>
> 4. **“Certainly, when the source and target share the same features, they become transferable. However, the reverse is not necessarily true.”**: We agree that from a mathematical point of view, as long as the transferred parameters are the same or similar, transfer learning is beneficial even if there are no shared features. However, in practice, many use cases of transfer learning still rely on the similarity of shared features. This is why we use the current setting that there exists a partial similarity between the source and the target tasks. We will revise the text to make these points clear.
>
> 5. **“The paper does not go beyond explaining the algebraic relationships.”**: The algebraic relationships are very important to understanding the fundamental properties of transfer learning and to the design of transfer learning strategies. Besides the algebraic relationships, we also provide intuitive reasonings, e.g., in Sec. 4.1, we give an intuitive comparison of Options A and B: *An intuitive explanation of the reason for these differences is that Option B does train the common part learned by the source task but Option A does not. Thus, Option B should do better to learn the common part. At the same time, since Option B uses more parameters ($p+p_{(2)}$) than Option A ($p$) to learn the target task's samples, the noise effect is spread among more parameters in Option B than in Option A, and thus Option B can mitigate the noise better than Option A. …*
>
>
>
>
>
> 6. **“find a subspace/transformation in which the source and target datasets share common features.”**: We agree that learning a feature representation shared across tasks itself is already an important research direction, e.g., [ref c]. Of course, generalizing our results to this setting will also be meaningful. Since our current analysis under known features is already very complicated, we plan to leave it to the future work. Moreover, such generalization of our results to the suggested setting will be able to build on and exploit our current analysis. We will revise the paper to comment on this important point.
>
> [c] Nilesh Tripuraneni, Michael Jordan, and Chi Jin. On the theory of transfer learning: The importance of task diversity. NeurIPS, 2020.

---

### Meta-Review · Area_Chair_SYWk · 2023-12-05

**Metareview:**

This paper addresses the problem of transfer learning for linear regression considering the presence of common and task-specific features. A theoretical analysis is provided both for the under-parameterized and over-parameterized scenarios accompanied with some numerical evaluation.

Based on reviewers' evaluation, the paper studies an important problem, the provided analysis studies interesting scenarios for feature analysis.
On the negative side, the necessity to have to know shared and specific features is seen as an important limitation (in addition to the linear setting but to a lesser extend), the presentation of the results and general clarity can be improved as well as the positioning with respect to SOTA papers.

During the rebuttal, authors have provided multiple answers to the points raised by the reviewers.
Some precision about the setting and the contribution have been appreciated.
The answers were not completely convincing, in particular with respect to the point related to the knowledge of shared and specific-features. References given by the authors address mainly contrastive learning or meta-learning, and the link with the contribution does not appear to be direct and does not appear sufficient to justify the setting.

The presentation of the contribution lacks of a better positioning and a larger and more precise positioning with respect to the state of the art. The answers provided in the rebuttal are a good start but they do not appear as sufficient making the contribution limited.

As a consequence I propose rejection.

**Justification For Why Not Higher Score:**

Limitations discussed in the additional comments appear to be too important.

**Justification For Why Not Lower Score:**

N/A

---

### Decision · Program_Chairs · 2024-01-16

Reject